# Cell-specific proteome analyses of human bone marrow reveal molecular features of age-dependent functional decline

Marco L. Hennrich [1,2], Natalie Romanov [1], Patrick Horn[2,3], Samira Jaeger[4], Volker Eckstein[3], Violetta Steeples[5], Fei Ye[1,2], Ximing Ding[1,2,3], Laura Poisa-Beiro[2,3], Mang Ching Lai[1,2], Benjamin Lang [1], Jacqueline Boultwood[5], Thomas Luft[3], Judith B. Zaugg [1,2], Andrea Pellagatti [5], Peer Bork [1,2,6], Patrick Aloy[4,7], Anne-Claude Gavin[1,2] & Anthony D. Ho [2,3]

Diminishing potential to replace damaged tissues is a hallmark for ageing of somatic stem cells, but the mechanisms remain elusive. Here, we present proteome-wide atlases of age-associated alterations in human haematopoietic stem and progenitor cells (HPCs) and five other cell populations that constitute the bone marrow niche. For each, the abundance of a large fraction of the ~12,000 proteins identified is assessed in 59 human subjects from different ages. As the HPCs become older, pathways in central carbon metabolism exhibit features reminiscent of the Warburg effect, where glycolytic intermediates are rerouted towards anabolism. Simultaneously, altered abundance of early regulators of HPC differentiation reveals a reduced functionality and a bias towards myeloid differentiation. Ageing causes alterations in the bone marrow niche too, and diminishes the functionality of the pathways involved in HPC homing. The data represent a valuable resource for further analyses, and for validation of knowledge gained from animal models.

[1] European Molecular Biology Laboratory (EMBL), Structural and Computational Biology Unit, Meyerhofstrasse 1, Heidelberg D69117, Germany. [2] Molecular Medicine Partnership Unit (MMPU), Meyerhofstrasse 1, Heidelberg D69117, Germany. [3] Department of Medicine V, Heidelberg University, Heidelberg D69120, Germany. [4] Institute for Research in Biomedicine (IRB Barcelona), The Barcelona Institute of Science and Technology, Barcelona,08028 Catalonia, Spain. [5] Radcliffe Department of Medicine, University of Oxford and Oxford BRC Haematology Theme, Oxford OX3 9DU, UK. [6] Department of Bioinformatics, Biocenter, University of Würzburg, Würzburg D97074, Germany. [7] Institució Catalana de Recerca i Estudis Avançats (ICREA), Barcelona, 08010 Catalonia, Spain. These authors contributed equally: Marco L. Hennrich, Natalie Romanov, Patrick Horn. Correspondence and requests for materials should be addressed to A.-C.G. (email: gavin@embl.de) or to A.D.H. (email: anthony_dick.ho@urz.uni-heidelberg.de)

Ageing of stem cells has been considered as the underlying cause for ageing of tissues and organs, especially in a biological system that is characterized by a high turnover such as haematopoiesis[1,2]. In humans, anaemia, decreased competence of the adaptive immune system, an expansion of myeloid cells at the expense of lymphopoiesis, and a higher frequency of haematologic malignancies have been reported to be hallmarks of ageing[3–5].

The age-associated phenotypes are initiated at the very top of the haematopoietic hierarchy, i.e., in the haematopoietic stem and progenitor cells (HPCs)[2,6]. With age, the HPC population undergo both quantitative (e.g., an increase in number) and functional changes (e.g., a decreased ability to repopulate the bone marrow[3,4,7,8]). Transcriptomic studies have provided a blueprint of the underlying molecular mechanisms and indicated that genes associated with cell cycle, myeloid lineage specification, as well as with myeloid malignancies were up-regulated in old HPCs, when compared to young ones[5,9,10]. The aforementioned knowledge on the various mechanistic aspects of HPC ageing was mostly, if not exclusively, gained by studies in murine models of ageing and has yet to be validated in human subjects.

Additionally, changes in the HPC microenvironment—the bone marrow niche—also influence haematological ageing. Whereas alterations in adhesion molecules, which are expressed in the cellular niche, and which are essential for homing and maintenance of HPCs, have been described, how they vary with the ageing process has not been defined[11–16]. In previous studies, we demonstrated specific transcriptomics and epigenetic alterations characteristic for ageing of human mesenchymal stem/stromal cells (MSCs)[17,18], while other groups indicated that different cellular elements in the marrow such as monocytes and macrophages could also play major roles[19–21]. Whereas these various mechanisms of ageing have been studied in a few, individual cell populations constituting the bone marrow, our understanding of the roles of intrinsic mechanisms, i.e., in the HPCs, vs. extrinsic ones, such as in the marrow niche, has remained fragmented.

The overarching goal of this study is therefore to acquire a systems understanding of the molecular mechanisms involved in ageing of human HPCs, as well as those in the cell populations comprising the marrow niche. As cell functions are more directly characterized by their proteins than their transcript complements, we performed a comprehensive and quantitative proteomics analysis of the HPCs and their niche in a large cohort of human subjects from different age groups. The underlying datasets should represent not only a valuable resource for mechanistic analyses and for validation of knowledge gained from animal models, but also provide an atlas of proteomic signatures of human ageing processes within the cellular network of the bone marrow. The systemic data should build a foundation for a better understanding of age-related diseases such as myelodysplastic syndromes (MDS) in the future.

## Results

### Multi-scale proteomics profiling of human bone marrow cells.
Bone marrow samples of high quality and sufficient quantity from 59 human subjects, 45 male and 14 female, were available for proteomics analysis (Fig. 1a, b). Their age ranged from 20 to 60 years with a median of 33.2 years, as depicted in Fig. 1b (Supplementary Table 1). From each bone marrow sample, we isolated HPCs as defined by CD34[+] and five other cell subpopulations, namely, lymphocytes and precursors (LYMs), monocytes/macrophages and precursors (MONs), granulocytic (GRAs), and erythroid (ERPs) precursors, as well as MSCs (Fig. 1a, b, Supplementary Fig. 1). The CD34[+] cells are highly enriched

mainly for progenitors, but do contain a significant percentage of stem cells. These cell populations constitute 94.2% (±2.8%) of all mononuclear cells in the bone marrow.

To simultaneously assess the molecular alterations associated with ageing at both spatial (cell population) and temporal (ageing) resolution, we combined two complementary mass spectrometry (MS)-based quantification methods. The cell type-specific changes were measured by a label-free technique adapted from Schwanhausser et al.[22] (see Methods) and the age-associated ones by isobaric labelling using the tandem mass tag (TMT)[23] (Supplementary Fig. 2a,b). The six different cell populations were analysed separately. Tryptic digests were processed in batches of five human subjects, each labelled with a different TMT, and were combined (Fig. 1c) with a TMT-labelled cell population-specific internal standard to ensure an accurate quantification across all samples. In total, we were able to identify 12,158 proteins (Fig. 1d, e; Supplementary Fig. 2b and Supplementary Data 1) (i.e., in all human subjects and in the six cell populations), which represents 77% of the currently detectable human proteome[24] (see Methods). The number of proteins characterized in each specific cell population varied from 6340 in ERPs to 9454 in MSCs.

The technical reproducibility for the TMT-based quantification, measured in triplicate, was high (averaged Pearson's correlation coefficient = 0.94; median coefficient of variation = 4.2%; details in Methods). A principal component analysis showed that age accounted for a small but sizeable fraction of the overall inter-individual variability (between 3.1 and 21.8% depending on the cell population). Based on these principal component analyses, we excluded samples that deviated significantly from the majority due to various technical issues (Supplementary Fig. 2c). After removing those outliers, we were able to analyse a total of 270 samples, and acquired 7375 protein abundance profiles across the human cohort (TMT quantification), and 6952 across the cell populations (label-free quantification). Although we could not capture the least abundant proteins[25], our analyses broadly cover the main functional and compartmental protein categories (Supplementary Fig. 3).

The quality of the dataset was validated by the inter-individual variability in each of the different cell populations studied (biological variability). While protein abundance fluctuated among the different samples within the same cell population, proteins within specific pathways or protein complexes that were expected to be co-expressed showed coherent changes across donors (Supplementary Fig. 4)[26,27]. Furthermore, hierarchical clustering based on all proteins quantified by label-free quantification yielded a distinct pattern permitting separation into clusters that recapitulate the known lineage relationship (Supplementary Fig. 5).

The quality of the dataset was further confirmed by the abundance profiles of known cell type-specific markers of the corresponding subpopulations (Fig. 2a and Supplementary Fig. 6), based on which the respective cell types were isolated (Supplementary Fig. 1). All validation analyses indicated that the proteomics datasets were reliable and of appropriate quality to address questions on age-dependent differences across cell populations.

### The proteomic landscape of the human marrow niche.
Recent human proteome atlases are still largely based on transcript abundances to estimate relative protein levels[24,25]. They also usually describe entire organs, and as a consequence the individual cell types that constitute the respective organs remain largely overlooked. Our data represent a comprehensive proteomics study that delineates the differences between the six major cell populations constituting the human bone marrow.

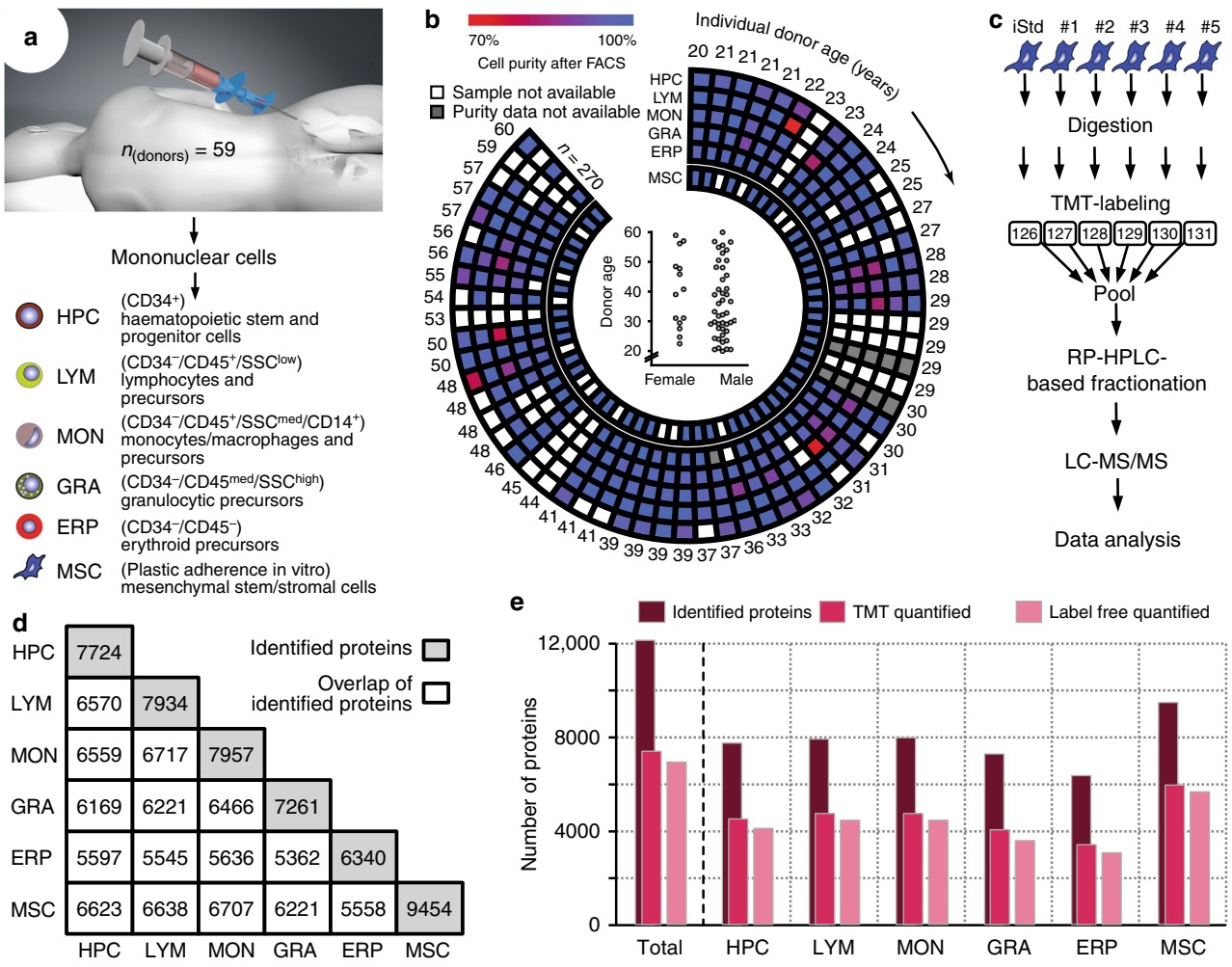

**Fig. 1** Summary of the experimental design of the proteomics study and description of the dataset. **a** Bone marrow samples were aspirated from human subjects and mononuclear cells were separated by FICOLL gradient centrifugation. Haematopoietic stem and progenitor cells (HPCs), lymphocytes and precursors (LYMs), monocytes/macrophages and precursors (MONs), granulocytic (GRAs), and erythroid precursors (ERPs) were isolated by fluorescence-activated cell sorting (FACS). Mesenchymal stem/stromal cells (MSCs) were expanded in cell culture. **b** The purity of the 270 samples derived from the 59 eligible human subjects is visualized by a colour gradient ranging from 70% (red) to 100% (blue). Each circle represents one cell type. The age of each of the individual donors is indicated at the periphery of the circle. The graph inside the circle depicts the age distribution of the 59 subjects separated by gender. **c** The different cell populations were processed separately for proteomics analysis. The samples of the same cell population from five human subjects, together with a cell type-specific internal standard, were processed in one batch. The cells of the samples were lysed, digested, labelled with the tandem mass tag (TMT), and subsequently pooled. This pool was separated by reversed phase HPLC and fractionated. The fractions were analysed by LC-MS/MS followed by bioinformatics analysis. **d** This graphic shows the pairwise overlap of the proteins identified in each cell population with the diagonal indicating the total number of proteins identified in the respective cell type. **e** Overview on the total number of proteins identified, quantified with TMT or by label-free quantification across all (total) as well as for the individual cell types

Only a fraction (8.3%; 578 proteins) of the proteome was expressed in a strictly cell-specific manner. Among these cell-specific proteins, some might be associated with specific functions of the respective cell type and serve as markers for isolation. For example, out of the 17 HPC-specific proteins, some are involved in the differentiation along the myeloid (dachshund homologue 1 (DACH1)) or lymphoid lineages (DNA nucleotidylexotransferase (DNTT); B-cell lymphoma/leukaemia, 11A (BCL11A); haematopoietic SH2 domain containing protein (HSH2D)), or in the maintenance of pluripotency (B-Box and SPRY domain containing protein (BSPRY))[28]. The restricted expression of these proteins in the HPC population, and their absence in the more committed progenitors, suggest a role in early stages of haematopoiesis.

When compared to HPCs and other haematopoietic progenitor cells, MSCs have the most distinct proteome with 452 proteins

uniquely expressed (7.7% of the quantified proteome). This reflects different biology and functional competences (see below), but also to some extent the fact that these cells had to be expanded in vitro because of their very low abundance in the bone marrow. Of these uniquely expressed proteins, 56 (12%) play a role in the organization of the extracellular matrix (ECM), and might contribute to MSC-mediated HPC homing[29]. Many of the proteins expressed uniquely in MSCs were localized at the cell surface, e.g., CD51 (ITGAV), CUB and LCCL domain containing 2 (DCBLD2), trophoblast glycoprotein (TPBG), discoidin (DDR2), the caveolea-associated proteins caveolin 1 (CAV1) and EH domain containing 2 (EDH2). They represent potential candidates for the characterization of MSCs as specific markers. Abundant and specific expression of nestin (NES) was found in human MSCs in the present study. The presence of nestin-positive MSCs has been reported to characterize a perivascular

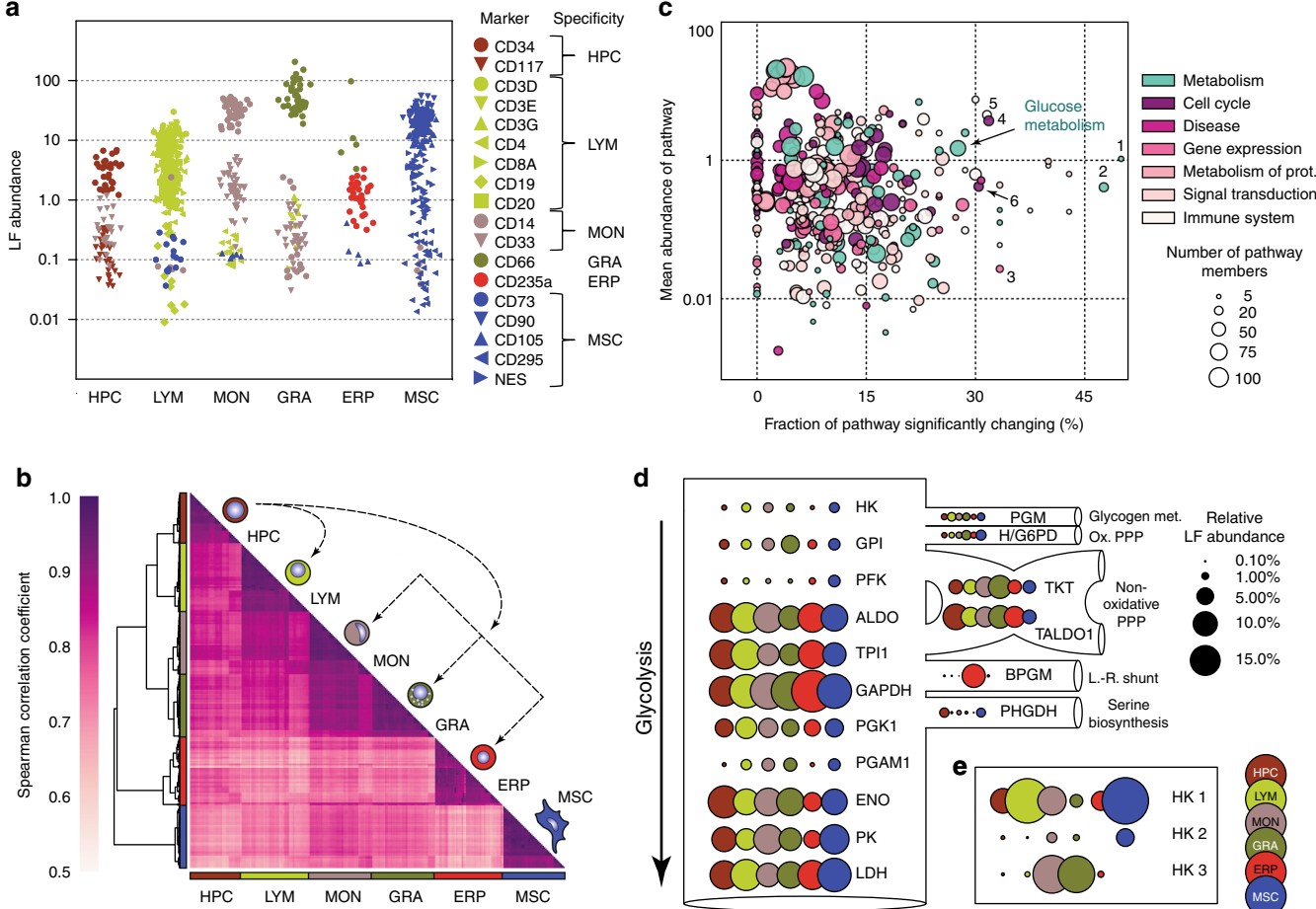

**Fig. 2** Quality of the dataset and differences between the cell populations. **a** Jitter plot displaying the relative label-free (LF) abundance of established cell type-specific markers. Each symbol represents an individual sample. **b** Correlation matrix of Spearman's rank correlation coefficients between all 270 samples and depiction of the haematopoietic subpopulations examined. The correlation coefficient was calculated based on the LF abundances of proteins covered in ≥85% of the samples of each cell population (950 proteins). Characteristic proteome clusters were identified in samples derived from the same cell population. **c** Bird's-eye view on pathways with different stoichiometry in the different cell populations. The fraction of a pathway that changed significantly between at least two cell types (x-axis) is plotted against the mean label-free (LF) abundance of an entire pathway across the different cell populations. The circles in the scatter plot signify Reactome pathways and their size corresponds to the number of quantified proteins per pathway. The colour refers to the highest hierarchy of the particular pathway according to Reactome. The highest hierarchies were filtered to have at least 30 pathways quantified in the dataset with each containing 5–100 protein members. The numbers indicate the following pathways; 1 ethanol/oxidation; 2 Histidine/Lysine/Phenylalanine/Tyrosine/Proline/Tryptophan catabolism; 3 RNA polymerase II transcription termination; 4 DNA replication pre-initiation; 5 endoplasmic reticulum-phagosome pathway; 6 activation of ATR in response to replication stress. **d** The glycolytic pathway and branching points to connected pathways, e.g., the oxidative pentose phosphate pathway (Ox. PPP) or the Luebering–Rapoport shunt (L.-R. shunt), are depicted. Each bubble represents an enzyme with the area of the bubble encoding for the cell type-specific, relative LF abundance with 100% representing the sum of all proteins per cell population depicted in **d**. The colour codes for the cell population as depicted in the lower right corner. **e** Relative abundances of the hexokinase isoenzymes in the respective subpopulations are shown analogous to **d**, but fivefold enlarged to better illustrate the difference

bone marrow niche, which supports HPC maintenance and homing in murine models[30,31] and in human bone marrow[32,33]. Our study has confirmed the existence of nestin-positive MSCs in adult marrow.

The vast majority (73.3%) of the quantified proteins were present in more than one cell population and, among those, 18.6% were consistently found in all six subpopulations (Supplementary Data 2). The abundance of those 950 commonly expressed proteins constituted up to ~70% of the quantified proteome (Supplementary Fig. 7). This core proteome was enriched in proteins with essential, housekeeping functions, but their abundance differed in the different cell populations. Especially, many metabolic pathways showed specific stoichiometries (Fig. 2c, Supplementary Data 3). Collectively, the relative stoichiometry of the commonly expressed proteins could be used

to define the six different cell populations (Fig. 2b). This might reflect their specific metabolic requirements for lineage commitment and adaptation to cell-specific processes and functions[27] (see below).

Prominent examples of such core proteins are those involved in the glycolytic pathway that converts glucose to pyruvate, and thereby provide both energy in the form of adenosine triphosphate (ATP) as well as carbon substrates for diverse anabolic processes. Whereas the relative abundance of these enzymes differed in the various cell populations, the cell-specific stoichiometry patterns were rigorously maintained across the different human donors (Supplementary Fig. 8). For example, when compared to other cell populations within this cellular network, ERPs were characterized by the most divergent enzyme stoichiometry (Fig. 2d). All enzymes downstream of the

glyceraldehyde 3-phosphate dehydrogenase (GAPDH) were less abundant when compared to all other cell types. This can be explained by the unique Luebering–Rapoport glycolytic shunt in erythrocytes that converts the 1,3-bisphosphoglycerate (the product of GAPDH) to 2,3-bisphosphoglycerate (2,3-BPG). The latter regulates oxygen release from haemoglobin and its delivery to tissues[34,35]. This indicated an early specialization of the glycolytic pathway for the production of 2,3-BPG in ERPs.

Similarly, we found a cell-specific expression of the different isozymes of hexokinase, the first rate-limiting enzyme in glucose metabolism (Fig. 2e). These isozymes control the metabolic fate of glucose-6 phosphate (G6P)[36]. While HPCs, LYMs, ERPs, and MSCs mainly expressed hexokinase 1, which channels G6P to glycolysis, both MONs and GRAs expressed significant amounts of hexokinase 3 that directs G6P to anabolic pathways, such as the pentose phosphate pathway. Consistent with the high level of hexokinase 3 in MONs and GRAs, enzymes in the oxidative and non-oxidative branches of the pentose phosphate pathway were also highly abundant in these subsets (Fig. 2d). This pathway generates nicotinamide adenine dinucleotide phosphate (NADPH)—and precursor for the synthesis of nucleotides—that play important roles in neutrophils, granulocytes, and macrophages[37].

Taken together, the dataset represents an important resource that not only depicts proteome adaptations to cell type-specific functions, but also quantifies the respective impact of biological variations in the different bone marrow subpopulations that reflect adaptation at the level of cell communities or tissues.

**Impact of ageing on proteome landscapes.** To quantify the proteomic alterations associated with human ageing, we performed Spearman's correlation analysis between the abundance of proteins consistently measured in >15% of all subjects and the corresponding chronologic ages (see Methods). Ageing was associated with subtle but significant changes in the abundance of many proteins ($p$ value ≤ 0.05, Spearman's correlation; Supplementary Data 4). These changes were not confounded by gender differences, as very similar values were obtained by excluding the 14 female subjects ($0.850 < r < 0.906$, Pearson's correlation coefficient) (Supplementary Fig. 9a). We have conducted transcriptomic analyses (using RNA-sequencing (RNA-seq)) on 65 samples (after fluorescence-activated cell sorting (FACS)) that were collected from the same human subjects and on the same day as the corresponding samples for proteomics analyses (Supplementary Fig. 9b). We observed that genes that were up-regulated at the protein level ($p$ value < 0.1, Mann–Whitney $U$-test) exhibited a significantly higher transcript fold change than genes that were down-regulated at the protein level (Supplementary Fig. 9b, c; Supplementary Data 5).

The age-associated alterations varied considerably in the different cell populations. For example, the number of age-related changes in protein abundance ranged from 175 (5.2%) in ERPs to 411 (9.1%) in HPCs and to 737 (12.4%) in MSCs (Supplementary Data 6). In addition, the different age-associated datasets were only partially overlapping, suggesting that age has distinct impacts on different cell types. This finding might reflect the varying half-lives of the six cell populations studied. HPCs and MSCs are relatively long-lived, persisting progenitor and stem cells, while the other cell types (LYMs, GRAs, MONs, and ERPs), with the exceptions of memory lymphocytes, represent lineage-committed precursors with higher turnover rates and considerably shorter half-lives.

To obtain an overview of the biological processes and pathways affected by ageing in each of the different subpopulations, we examined the systems alterations by combined analysis of all proteins involved in specific pathways defined in the Reactome database (http://www.reactome.org) (Fig. 3a, Supplementary Fig. 10 and Supplementary Data 7). This analysis revealed remarkable cell type-specific and age-associated changes in protein abundances. For example, we detected a significant increase in glycogen breakdown, synthesis of prostaglandins and thromboxanes (arachidonic acid metabolism), and metabolism of nitric oxide in older HPCs (see below). In the other cell populations, these processes were largely unaffected. In older MSCs, prominent alterations included differential regulation of proteins that are associated with cellular response to stress, replicative senescence, white adipocyte differentiation, and ECM organization.

Our data captured some of the few established ageing markers identified by transcriptomic analyses. For example, there was a reduction in abundance of interferon regulatory factor 8 (IRF8) as the HPCs become older, a phenomenon reported to be associated with dysregulated proliferative activity and biased myeloid differentiation[38] (Fig. 3b). The mitochondria play an important role in ageing[39] and we identified increases in mitochondrial calcium uniporter (MCU), ATP synthase, $H^+$ transporting mitochondrial F0 complex subunit G (ATP5L), and a significant decrease in mitochondrial ribosomal protein L48 (MRPL48) upon ageing. There was also a significant reduction in abundance of DNA methyltransferase 1 (DNMT1), a protein that is responsible for maintenance of DNA methylation pattern[40] (Fig. 3b). In human systems, its expression is frequently down-regulated in most common age-related diseases[41] such as acute myeloid leukaemia) and MDS. In mouse models, Dnmt1 is critical for HPC maintenance and their ability to self-renew efficiently after transplantation[42].

**Ageing affects central carbon metabolism in HPCs.** The most remarkable changes in proteome landscapes of HPCs associated with ageing were found in enzymes that play a central role in glycolysis, glycogen catabolism, and fatty acid beta-oxidation (FAO). These alterations all indicated an enhanced metabolic and specifically anabolic activity of old HPCs vs. young HPCs (Fig. 4). In contrast, these changes were not found in the other five cell populations even though many of these proteins belong to their core proteome (Supplementary Fig. 11).

We identified a significant age-associated increase in abundance of enzymes catalysing the rate-limiting steps of the upper part of glycolysis, namely hexokinase 1 (HK1) and phosphofructokinase M (PFKM), as well as the glycolytic enzymes aldolase C (ALDOC) and triosephosphate isomerase 1 (TPI1) (Fig. 4a–d). Simultaneously, the main enzymes involved in glycogen catabolism, glycogen phosphorylases brain and liver form (PYGB, PYGL), and glycogen debranching enzyme (AGL) were significantly more abundant in older HPCs. Increased abundance of phosphoglucomutase 1 (PGM1) also indicated an enhanced propensity to fuel the upper part of the glycolysis with the catalytic products of glycogen. Compatible with these changes, we demonstrated an increase in the abundance of transaldolase 1 (TALDO1), a key enzyme of the non-oxidative pentose phosphate pathway. There was also an increase in the abundance of the glycerol-3-phosphate dehydrogenase (GPD2), and dihydroxyacetone kinase (DAK), thus rendering the full picture of increased activity in the upper part of the central carbon metabolism in ageing HPCs complete.

Notably, these changes affected only the preparatory phase of the glycolytic pathway that consumes ATP and converts glucose to dihydroxyacetone phosphate and D-glyceraldehyde 3-phosphate. The second phase—characterized by the production of ATP, nicotinamide adenine dinucleotide (NADH) and pyruvate

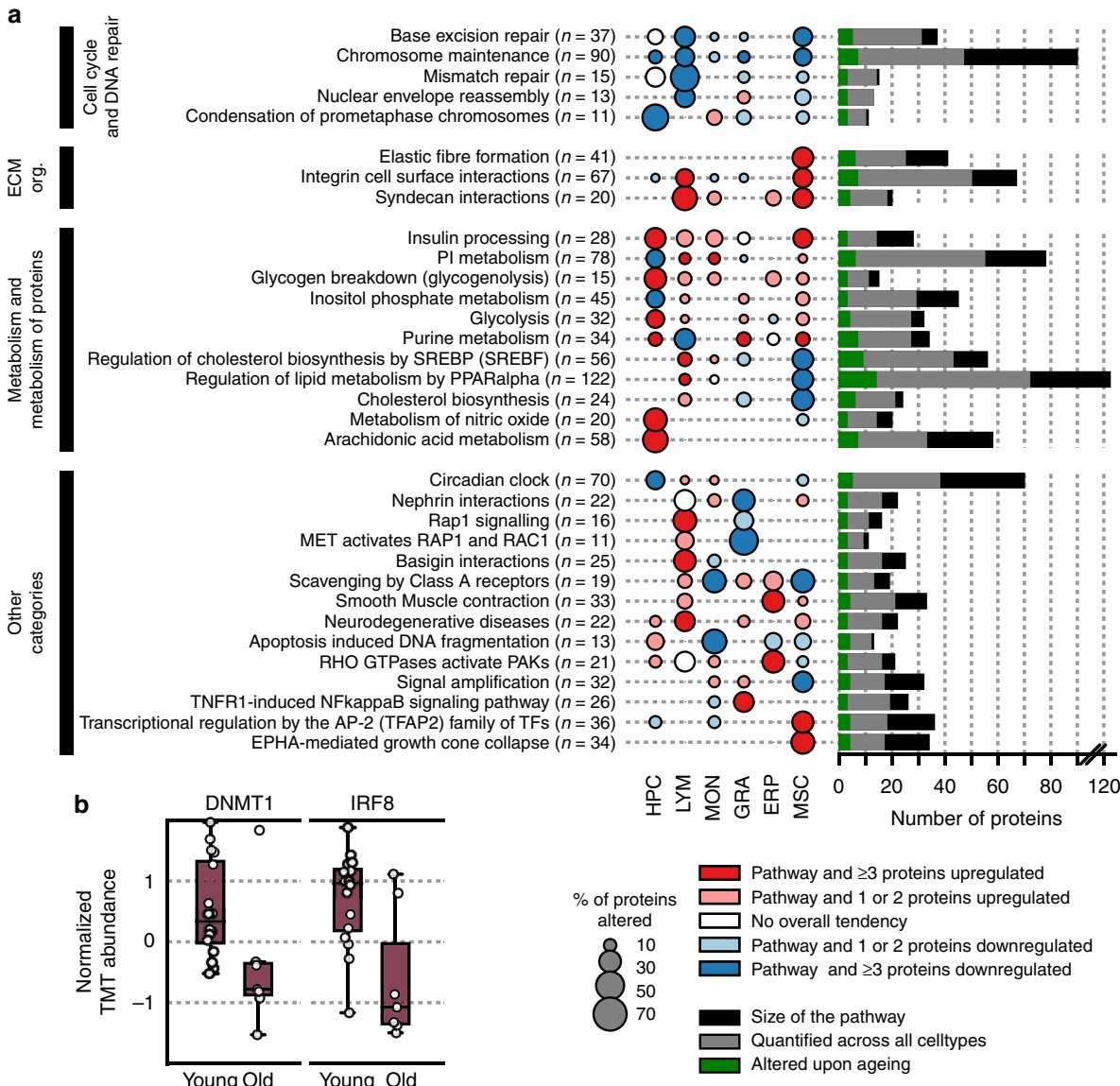

**Fig. 3** Age-affected pathways in the individual cell populations. **a** A selection of pathways from the Reactome database that show the most prominent changes upon ageing are depicted. Pathways were required to have between 5 and 150 members to be sufficiently covered in at least one cell population (>30% of the proteins are quantified), and to have at least 20% of its quantified components being significantly ($p$ value < 0.05, Spearman's correlation) altered upon ageing. The figure illustrates the five most up- and down-regulated pathways for each cell population. The area of the bubbles represents the percentage of proteins quantified by TMT that are significantly ($p$ value < 0.05, Spearman's correlation) altered. If no bubble is shown, no protein of the pathway was significantly altered in the respective cell population or no protein of the pathway has been quantified. Red indicates an overall increase of the pathway members with at least three proteins being up-regulated and pink indicates an overall increase with one or two proteins being up-regulated. Blue codes for pathways with an overall trend towards downregulation, with dark blue coding for pathways with at least three proteins being down-regulated and light blue containing one or two proteins being down-regulated. The colour white indicates that no overall tendency could be observed. The bars on the right-hand side of each pathway illustrate the number of proteins being significantly altered upon ageing regardless of the cell population (green), being quantified by TMT (grey), and the total number of members of the pathway (black) as also mentioned in the pathway annotation ($n$). The grouping of the pathways on the left side is based on the highest hierarchy levels defined in Reactome, e.g., extracellular matrix organization (ECM org.). **b** Effect of ageing on the abundance of DNMT1 and IRF8 in HPCs. The dots represent the individual results from young (age < 30 years) and old (age > 50 years) human subjects. The central line in the box plots indicates the median, the bottom and top edges of the box the interquartile range (IQR), and the box plot whiskers represent 1.5 times the IQR

—as well as the Krebs cycle remained largely unaffected (Fig. 4a). This is reminiscent of the Warburg effect, where excess of glycolytic carbon is redirected to pathways that branch out of the glycolysis/Krebs cycle axes, thus producing cofactors and intermediates for anabolism (e.g., nucleotides, lipids, amino acids synthesis)[43] and epigenetic processes[44]. Consistent with this trend towards anabolism upon ageing, metabolomics analyses of HPCs

($n = 10$, age: 21–69-year-old) revealed a tendency for two metabolites in the pentose phosphate pathway, ribulose 5-phosphate and ribose 5-phosphate, to accumulate in aged HPCs (Fig. 4b, Supplementary Data 8). In addition, phosphoglycerate dehydrogenase (PHGDH), an enzyme which is often over-expressed in tumours[45] and that diverts 3-phospho-D-glycerate out of the glycolytic pathway to convert it to 3-

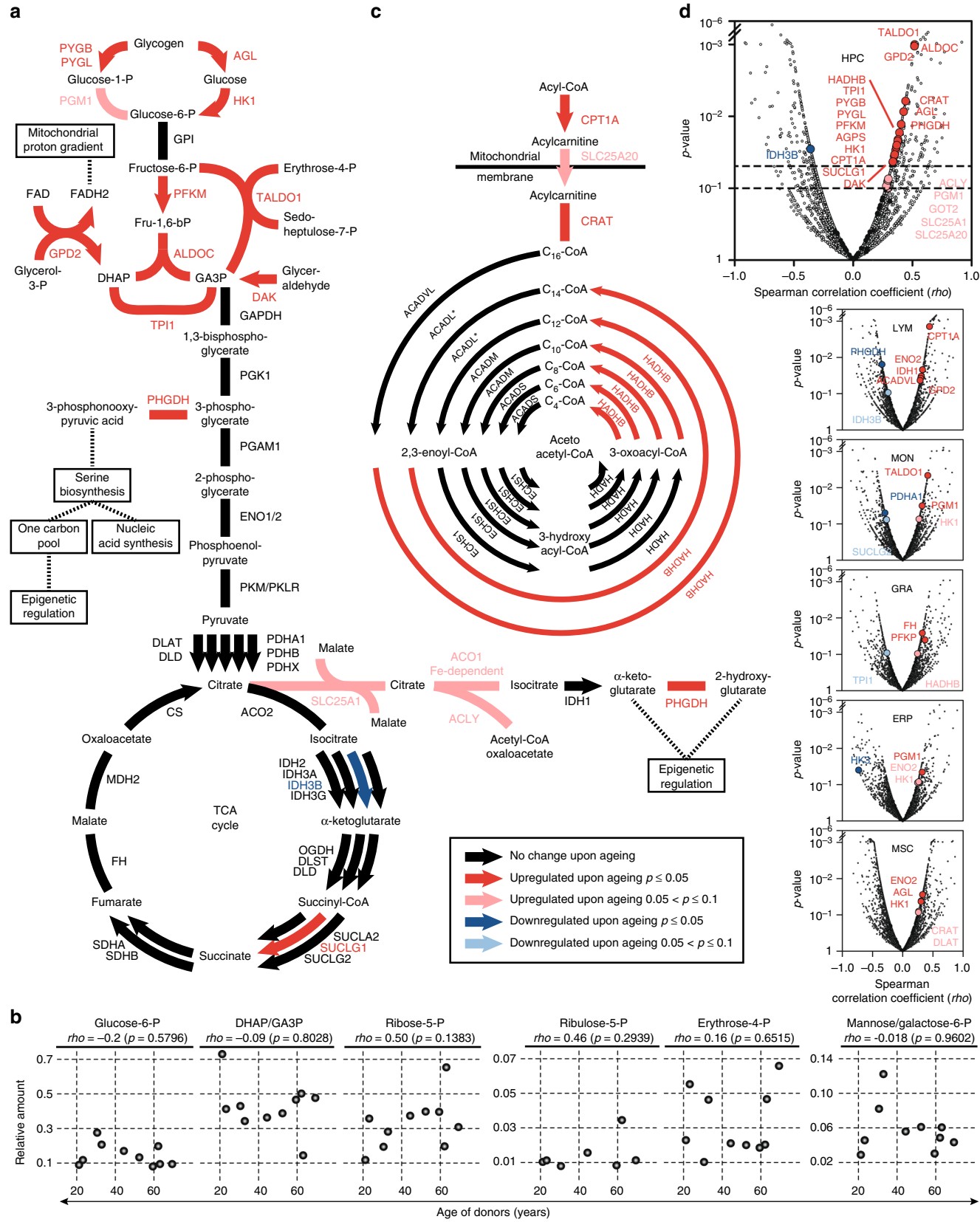

phosphonooxypyruvate for the biosynthesis of serine, increased in abundance in older HPCs. In cancer cells, a high demand for serine is known to support nucleotide synthesis, NADPH production, and the biosynthesis of *S*-adenosyl methionine (SAM), a methyl donor for essential biochemical processes[43,46].

Likewise, the mitochondrial citric acid transporter SLC25A1 that shuttles citrate to the extra-mitochondrial periphery increased in abundance with age along with an increased abundance of soluble aconitase (ACO1) and ATP citrate lyase (ACLY). Both enzymes use citrate as a substrate outside of the

**Fig. 4** Prominent changes upon ageing in the central carbon metabolism. **a** The glucose metabolism and the tricarboxylic acid (TCA) cycle of HPCs are depicted, with arrows representing unidirectional reactions and strokes representing bidirectional reactions. The gene names of the respective enzymes are written in capital letters and the colour codes for changes upon ageing, as described in the legend. A star indicates that the particular protein was not covered for quantification. Apart from the main glycolytic pathway, glycogen breakdown as well as a part of the pentose phosphate pathway is shown. **b** Measurements of relative amounts of phosphorylated metabolites relevant to the preparatory phase of the glycolytic pathway and the pentose phosphate pathway. The age of the respective donor (x-axis) is plotted against the relative amount of metabolites (y-axis) after normalizing for the cumulative amount of the given metabolites detected in each donor. Spearman's rank correlation coefficient is indicated as *rho*. **c** Similar to **a**, the scheme depicts the effects of ageing on a specific set of enzymes involved in the mitochondrial beta-oxidation of fatty acids. **d** Volcano plots of all proteins quantified in the respective cell populations. The dashed lines indicate p values of 0.1 and 0.05. Proteins represented in **a** and **c** are colour coded according to the legend. All other proteins are coloured grey. **a–d** The p values (p) are based on Spearman's correlation analyses

mitochondria. Synthesizing acetyl-CoA and oxaloacetate from citrate and CoA, ACLY is the key regulator between aerobic glycolysis and amino acid as well as de novo lipid synthesis involved in proliferation of tumour cells[47].

Simultaneously, there was also an age-dependent increase in abundance of FAO enzymes such as the trifunctional enzyme subunit beta (HADHB), peroxisomal bifunctional enzyme (EHHADH), propionyl-CoA carboxylase beta chain (PCCB), carnitine O-acetyltransferase (CRAT), as well as carnitine palmitoyl transferase 1 (CPT1A) and the carnitine-acyl-carnitine transporter SLC25A20 in aged HPCs (Fig. 4c). These enzymes span the whole fatty acid import machinery from the conversion of acyl-CoA to acyl-carnitine (CPT1A) in the cytosol, the transport of acyl-carnitine into mitochondria via SLC25A20 to the reconversion to acyl-CoA by carnitine o-acetyltransferase (CRAT)[48], and include major enzymes of mitochondrial (HADHB) and peroxisomal (EHHADH) FAO. All these changes were indicative of increased FAO as the HPCs became older. Together with glycolytic catabolism, FAO has been reported to be an important hallmark of HPC maintenance and quiescence[49,50]. Overall, these data support the notion that in HPCs, ageing is associated with a rewiring of central metabolic pathways and the rerouting of metabolic intermediates for the synthesis of cofactors important for anabolic and epigenetic processes.

**Higher myeloid versus lymphoid differentiation upon ageing.** Haematopoietic stem cells are characterized by their ability to both self-renew and differentiate into all functional blood cells. As mentioned above, 17 proteins were specifically found in the HPC population that are involved in the maintenance of pluripotency, or in the differentiation along the myeloid or lymphoid lineages. To characterize the age-associated functional attenuation of HPCs, we then examined the dynamics of alterations in abundance of these proteins with age (Fig. 5a). DNTT and BCL11A, two proteins directly linked to lymphoid development and function, decreased significantly as the HPCs became older. In contrast, the heterodimeric soluble guanylate cyclase (GUCY1A3 and GUCY1B3), i.e., downstream signalling effectors of nitric oxide (NO), increased significantly in abundance in older HPCs. NO/cyclic guanosine monophosphate (cGMP) signalling has been shown to modulate haematopoiesis and might also indicate an increased differentiation bias towards the myeloid lineage[51] (see below).

We further assessed the age-associated effects on lineage-relevant proteins by integrating known markers. Lymphoid marker proteins, such as MME (membrane metallo-endopeptidase, also known as CD10), IKZF1 (ikaros family zinc finger 1) and EBF1 (early b-cell factor-1) decreased with age. Furthermore, several proteins derived from a gene set characteristic for human lymphoid development[10] diminished in abundance with age as well. Figure 5b shows the alterations in abundance of HPC proteins that were reported to be relevant for lymphoid differentiation.

In sharp contrast, proteins that are associated with myeloid lineages, i.e., prostaglandin-endoperoxide synthase 1 (PTGS1), proline-serine-threonine phosphatase interacting protein 2 (PSTPIP2), thromboxane A synthase 1 (TBXAS1) that is associated with platelet development and function, as well as the promyelocytic leukaemia protein, all increased significantly in the older HPCs (Fig. 5c). In summary, we have demonstrated a significant decrease in abundance of proteins involved in lymphoid development, while factors associated with myeloid and platelet differentiation increased in abundance as the HPCs became older.

In order to examine whether the increased abundance of enzymes of the preparatory phase of the glycolytic pathway might be a direct consequence of the lineage skewing of the CD34+ cells towards myeloid differentiation, we analysed the transcriptomes of 519 single-cell sorted HPCs originating from young (n = 2) and old (n = 2) human subjects. Based on the abundance levels of the messenger RNA (mRNA) markers of lymphoid or myeloid differentiation, we categorized each individual HPC cell (Fig. 6a–c). The mRNA levels of age-increased glycolytic enzymes were higher in myeloid-primed than in lymphoid-primed HPCs, whereas transcripts for age-unaffected enzymes remained at similar levels in both subsets (Fig. 7a, b and Supplementary Fig. 12a,b). The age-dependent increase in glycolytic enzymes was most prominent in the myeloid-primed subset of HPCs. Thus, the lineage skewing of the CD34+ cells towards myeloid differentiation upon ageing may account, at least in part, for the increase in abundance of glycolytic enzymes.

**Changes in the bone marrow niche in relationship to HPCs.** The bone marrow represents a specialized environment that controls HPC maintenance and regulates haematopoietic homeostasis. Our systematic approach has provided a unique opportunity to simultaneously define the age-dependent changes in the cellular components constituting the niche. Several essential factors and adhesion molecules produced by the cellular niche and responsible for homing and egress of HPCs[52–54], e.g., stromal cell-derived factor-1 (SDF-1/CXCL12), vascular cell adhesion molecule 1 (VCAM1), and fibronectin (FN1)[55], all decreased in abundance in older MSCs (Fig. 8). Simultaneously, there were significant changes in abundance of proteins tightly linked to glycosaminoglycan metabolism (galactosylgalactosylxylosylprotein 3-beta-glucuronosyltransferase 3 (B3GAT3); chondroitin sulphate N-acetylgalactosaminyltransferase 2 (CSGALNACT2); biglycan (BGN); alpha-L-iduronidase (IDUA)), and collagen metabolism (collagens: COL4A2, COL1A1, COL1A2, COL3A1, COL5A2, COL6A3, COL11A1; prolyl 4-hydroxylase subunit beta (P4HB)) as MSCs became older. These alterations indicated that ageing was associated with a reorganization of the ECM, as well as structural changes in the architecture of the bone marrow niche.

Ageing is probably associated with coordinated, concerted alterations in this network of cell communities within human bone marrow. To visualize the coordinated and simultaneous

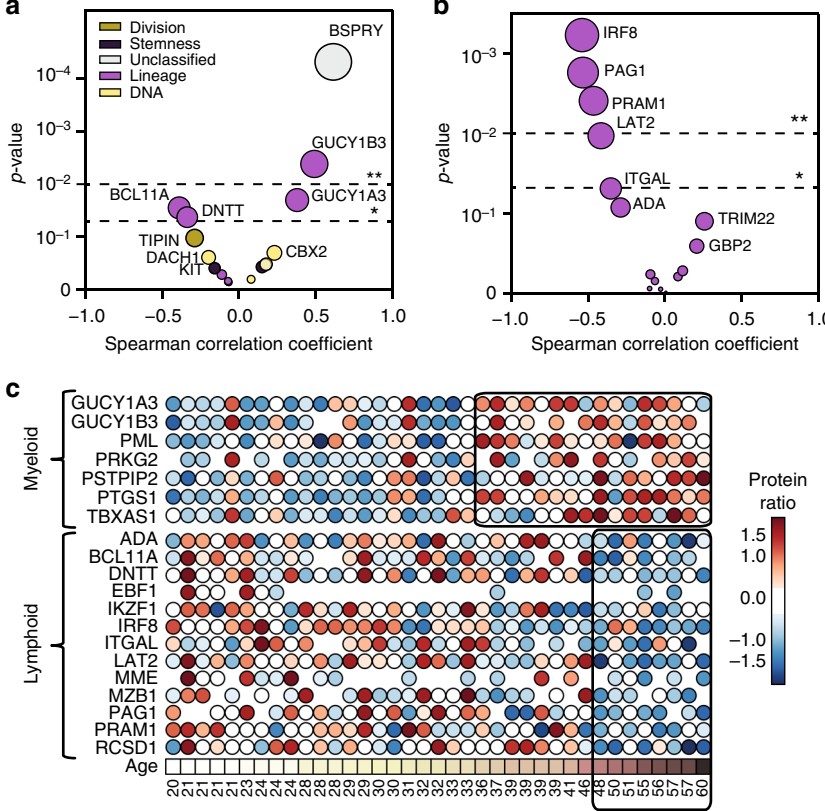

**Fig. 5** Age-related alterations of lineage-specific proteins in HPCs. **a** Scatter plot of the relationship between Spearman's rank correlation coefficient (x-axis) and the corresponding p value of proteins specifically expressed in HPCs (y-axis). The colour assigned to each bubble is indicative of its function, as explained in the upper left legend. The area of the bubbles represents the significance level (p value (−log10)). The stars at the dashed lines indicate a p value of 0.05 (*) and 0.01 (**). **b** Scatter plot similar to **a**, with proteins associated with common lymphoid progenitors (CLP) compared to megakaryocytic and erythroid progenitors (MEP), common myeloid progenitors (CMP), and haematopoietic stem cells/multipotent progenitors (HSC/MPP) in accordance with the GO annotation of immune system process (GO:0002376)[10]. **c** Significant alterations of the proteome landscape of HPCs upon ageing. Each circle represents a successful quantification in an individual human subject, with the respective age noted at the bottom. The colour codes for the z-score of relative intensity vs. the internal standard (TMT protein ratio), with red indicating an increase and blue a decrease in abundance upon ageing. The boxes indicate the age boundaries where the majority of the respective age-dependent increase or decrease in protein abundance takes place

changes that occur across the different subpopulations, we have designed a correlation matrix for all extracellular protein–ligand pairs measured in different cell populations[56] (Fig. 9). A correlative analysis of protein expression profiles within this context might be indicative of a direct or indirect functional relationship. Based on the STRING database[57], we demonstrated that 28% of these receptor–ligand pairs directly interacted with one another (p value = 0.0498, Fisher's exact test), while 62% showed indirect functional relations (p value = 0.047, Fisher's exact test). For example, complementary to the decrease in VCAM1 and FN1 described above in MSCs, their corresponding ligand, α4/β1 integrin (ITGA4/ITGB1), also decreased in HPCs upon ageing. Interestingly, αL/β2 integrin (ITGAL/ITGB2) followed a very similar pattern (Fig. 8). These observations supported the notion that β2-containing integrins on HPCs showed a synergistic effect with that of α4/β1 integrin, as reported in the literature[58].

Soluble factors secreted by the cellular components of the niche were also affected as the marrow niche became older. Some of these changes might account for the functional attenuation of HPCs described in the literature. For example, transforming growth factor beta-1 (TGFB1) was elevated in LYMs and ERPs from older subjects (Fig. 8b). In mouse models, TGFB1 has been proposed to contribute to lineage skewing by stimulating myeloid-biased HPCs, while inhibiting lymphoid

development[52,59]. Furthermore, we have found a significant decrease in abundance of the NO synthase inhibitor NOSIP in MSCs. This was associated with a prominent increase in abundance of dimethylargininase-1 and 02 (DDAH1 and DDAH2), enzymes that degrade asymmetric dimethylarginine (ADMA)—an inhibitor of NO synthase—in HPCs (Fig. 8b). These complementary changes in the HPCs suggested that the immunomodulatory secondary messenger NO was up-regulated in the ageing niche, whereas the downstream signalling effectors of NO, i.e., the heterodimeric soluble guanylate cyclase (GUCY1A3 and GUCY1B3) and the cGMP-dependent kinase 2 (PRKG2), were significantly elevated in older HPCs (Fig. 8b). Elevated levels of TGFB1 and NO in the bone marrow niche might represent initial triggers for the lineage skewing described above. Overall, these alterations have provided evidence that, during the ageing process, extrinsic factors in the niche play a major, complementary role to intrinsic factors in the HPCs.

## Discussion

We have presented an atlas of the age-associated alterations in proteomic landscapes of human HPCs as well as of five other subpopulations comprising the bone marrow niche. This comprehensive and comparative proteomics study of constituent cell types in a human tissue across the time span of 40 years is unique.

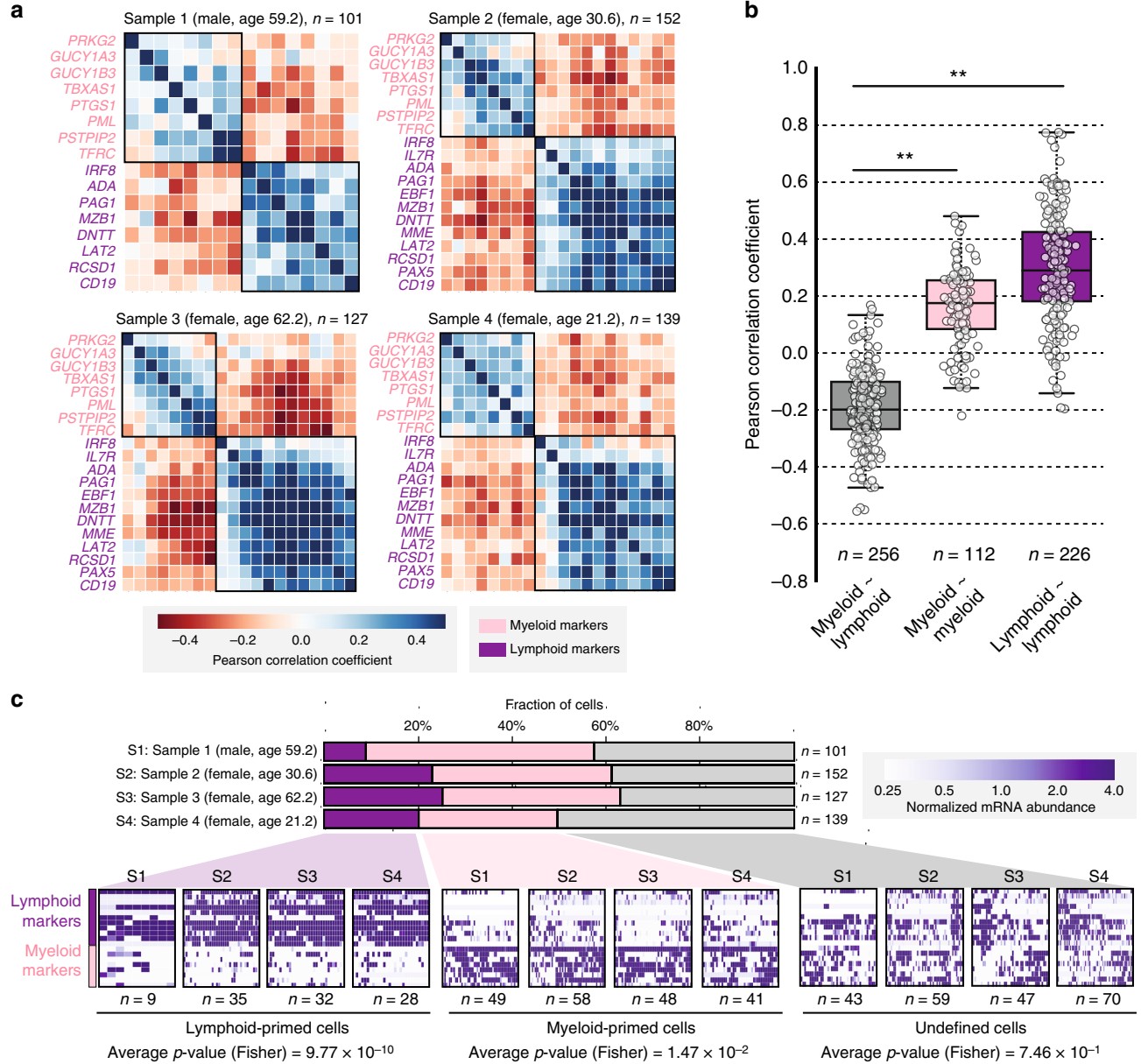

**Fig. 6** Classification of single CD34+ cells into myeloid- and lymphoid-primed cells. **a** The matrices show the correlation (Pearson) of the abundance of the mRNA markers specific for lymphoid or myeloid differentiation (see also Fig. 5c) in single HPCs. **b** The distribution of correlation values (Pearson) between myeloid and lymphoid markers (grey), between myeloid markers only (pink), and between lymphoid markers only (purple) is presented as box plot. The central line in the box plots indicates the median, the bottom and top edges of the box the IQR, and the box plot whiskers represent 1.5 times the IQR. Correlation values are taken from all four samples. Significance was assessed using Mann–Whitney U-test (**p value < 0.01). **c** Cells are classified into lymphoid- and myeloid-primed cells based on markers. The bar-plot gives an overview on the fraction size of lymphoid-primed (purple), myeloid-primed (pink), and undefined (grey) cells in a given sample (undefined meaning that cells could not be classified as myeloid or lymphoid). For each fraction and each sample (S1–S4) the heat maps show the mRNA abundance of lymphoid and myeloid markers. The significance of marker distributions in each fraction was assessed by Fisher's exact test (p values depicted)

Overall, our dataset provides a framework which allows for the interpretation and integration of the fragmented knowledge derived from various experimental models[6].

Among the significant alterations in terms of abundance of proteins upon ageing, the most prominent changes included an enhanced metabolic and anabolic activity of older HPCs as compared to young HPCs. Glucose metabolism has been shown to influence chromatin structure and transcription[43] on the one hand, and to play a pivotal role in governing stem cell fate in terms of proliferation, differentiation, or dormancy on the other[49]. Our study has provided evidence for a prominent shift in central carbon metabolism of human HPCs, indicating enhanced metabolic and anabolic activity in the HPCs during the ageing process. These changes are reminiscent of a Warburg effect[43,44].

Higher proliferation of aged HPCs is associated with a general loss of function, including a diminished regenerative potential in serial transplantation assays[6]. For example, murine models have provided convincing evidence for a decreased competence of the adaptive immune system, an expansion of myeloid cells[6], and an increased platelet priming and functional platelet bias[60] as the HPCs age. In humans, Pang et al.[9] observed that HPCs increased in frequency with age, but were less quiescent, and exhibited

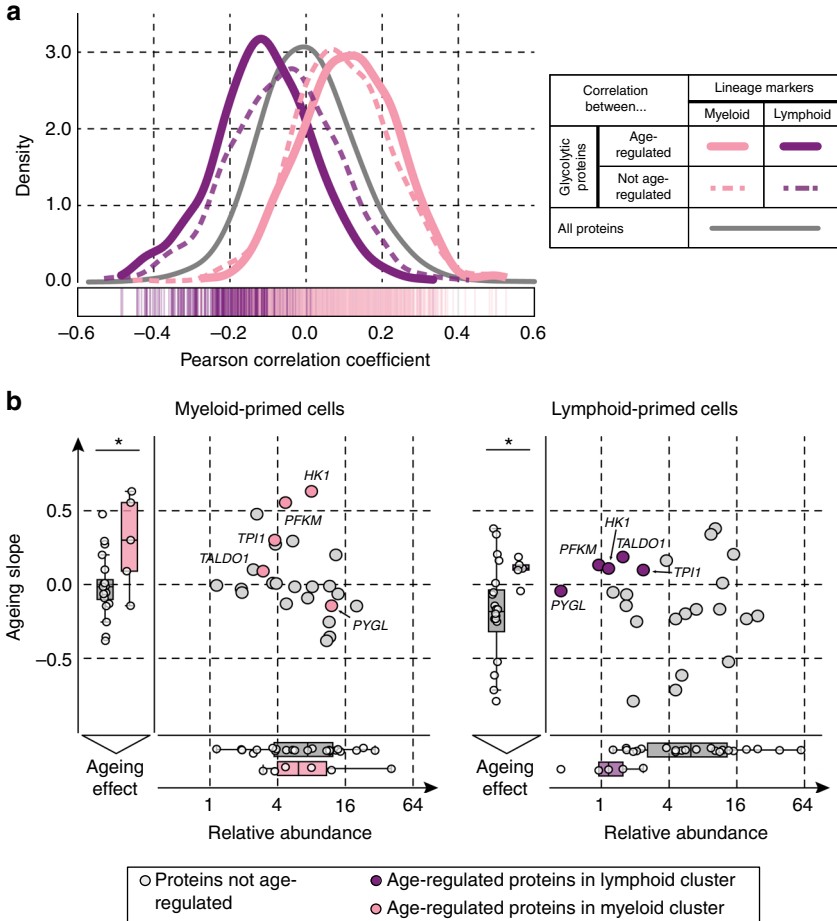

**Fig. 7** Single-cell analysis reveals lineage- and age-dependent increase of glycolytic enzymes. **a** The density plot shows distributions of correlation coefficients (Pearson) between lineage markers and glycolytic enzymes. The calculation is based on the results from the single-cell RNA-seq data. Density distributions based on correlations between myeloid (pink) or lymphoid (purple) markers with glycolytic enzymes that were up-regulated upon ageing, are shown with thick lines. The corresponding individual correlation values are displayed as lines in the box below the plot in the respective colours. Density distributions based on correlations of the respective markers with glycolytic and TCA-related proteins that did not change upon ageing are shown as dashed coloured lines. The grey distribution represents the correlation values of all proteins against myeloid and lymphoid markers. **b** Scatter plot illustrating the effect of ageing on glycolytic enzymes in lymphoid- and myeloid-primed cells, as deduced from single-cell analysis. The dots correspond to proteins in glycolysis that were not affected by age (grey), and glycolytic proteins that were altered upon ageing according to the proteomics data. The x-axis illustrates the relative abundance of those enzymes in myeloid-primed cells (left), and lymphoid-primed cells (right). The y-axis corresponds to the ageing slope derived as a measure of age-effect in lymphoid- and myeloid-primed cells across donors *$p$ value < 0.05, Mann–Whitney U-test (see also Supplementary Fig. 12a, b). The data from the scatter plots are collapsed into box plots on both axes with the central mark indicating the median, the bottom and top edges of the box indicating the IQR. The box plot whiskers represent 1.5 times the IQR

myeloid-biased differentiation potential. Thus, our data have provided a complete atlas of dynamic changes in intrinsic and extrinsic factors upon ageing and hence leverage for our strategy in profiling all the different compartments of the marrow niche.

The simultaneous investigation of the other five cell populations in the bone marrow constituting the HPC niche represents another uniqueness of our present study and revealed age-associated alterations in the interplay of the niche components. Several of the key factors responsible for homing, egress, and differentiation of HPCs (SDF-1/CXCL12, VCAM1, FN1, integrins α4, αL, β1, and β2) decreased in abundance, whereas soluble factors responsible for HPC differentiation[61,62] like TGFB1 increased in abundance in the niche with age. In addition, numerous alterations in older MSCs supported the notion that ageing is associated with changes in the ECM and in the architecture of the bone marrow niche. Overall, these changes might explain the previously described deficit of homing potential of

aged murine HPCs[54] and the reduced rate of homing of elderly recipients of bone marrow transplantations in murine models. Our data on dynamic changes in intrinsic and extrinsic factors upon ageing serve as an atlas that can be leveraged for profiling all the different compartments of human bone marrow.

We have captured the proteomics signatures of the ageing process in a human tissue and are able to present an atlas of comprehensive, age-related alterations in proteome landscapes of human HPCs and the cellular niche elements. Our datasets also represent a valuable resource and basis for development of treatment strategies targeting metabolic alterations and pharmacologic manipulations to enhance HPC regeneration.

## Methods
**Specimen and donor cohort**. Bone marrow samples were harvested from human subjects through puncture at the posterior iliac crest using a Yamshidi needle, with aspirations at 5 to 7 different levels of approximately 10 ml at each level[18]. The

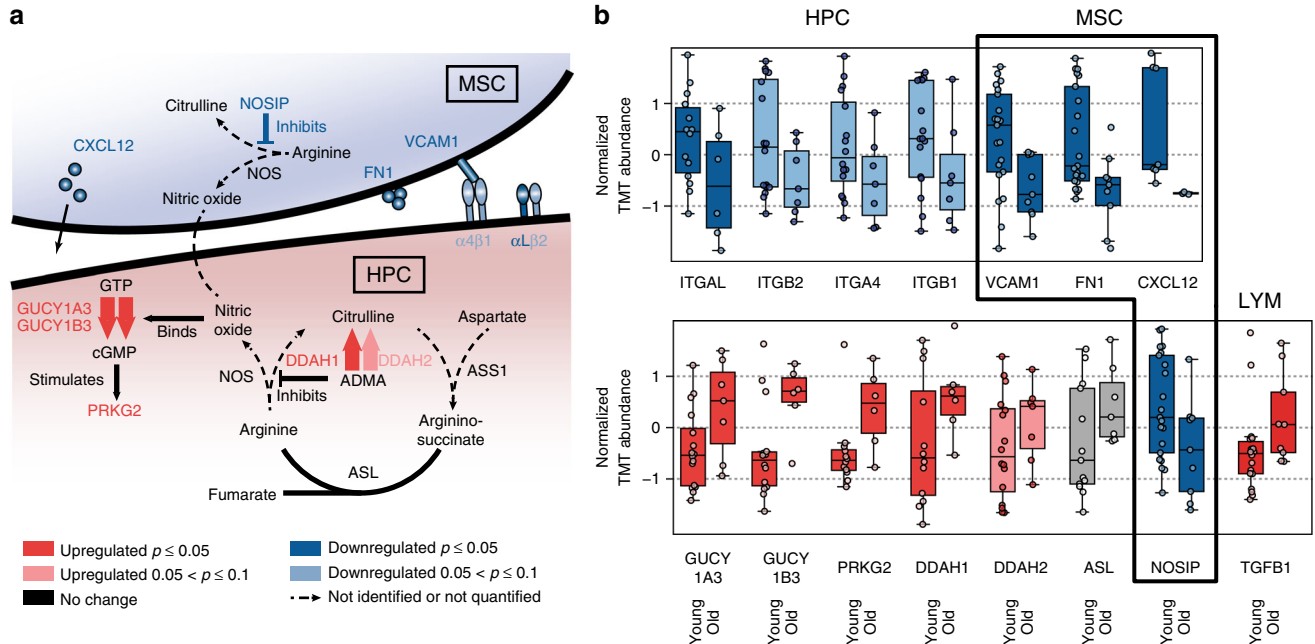

**Fig. 8** Alterations of protein abundance in the haematopoietic stem cell niche with age. **a** CXCL12, VCAM1, and FN1 in MSCs and the integrins alpha4, alphaL, beta1, and beta2 (α4, αL, β1, and β2) in HPCs decrease in abundance upon ageing. Age-related changes in connection with nitric oxide (NO) synthesis, the urea cycle, and potential NO crosstalk between MSCs and HPCs is depicted by arrows for unidirectional reactions and strokes for bidirectional reactions. Dotted lines or dotted arrows illustrate reactions for which no enzyme was detected or quantified in this study. The gene names of the respective enzymes are written in capital letters. The colour encodes changes upon ageing, as described in the legend (Spearman's correlation). **b** Box plot representation of the proteins depicted in **a**. The dots represent the individual results from younger (age < 30 years) and older (age > 50 years) human subjects. For all box plots, the central line indicates the median, the bottom and top edges of the box the IQR, and the box plot whiskers represent 1.5 times the IQR. Proteins known to have a direct effect on homing or egress of HPCs from or into the bone marrow are shown in the upper and proteins associated with NO signalling and the urea cycle are plotted in the lower graph. An additional box plot depicts the ageing effect on TGFB1 in lymphocytes

study has been approved by the Ethics Committee for Human Subjects at the University of Heidelberg, and written informed consent was obtained from each individual. We recruited 59 subjects, with the age ranging from 20 to 60 years for proteomics and transcriptomics studies. Gender and age distribution of the human subjects is depicted in Fig. 1b and listed in Supplementary Table 1.

For validation experiments (metabolomics and single-cell RNA-seq) 10 additional subjects were recruited for isolation and analysis of CD34+ cells (Supplementary Table 2). Mononuclear cells derived from umbilical cord blood were used as internal standards for the proteomics analyses.

**Cell isolation and sample generation.** The aforementioned bone marrow aspirates were processed by FICOLL density fractionation for isolation of mononuclear cells (MNCs). After staining with CD34-APC, CD45-FITC, and CD14-PE (all from BD Biosciences, San Jose, CA), five different cell populations were isolated as indicated in Fig. 1a and Supplementary Fig. 1 using a FACSAria II flow cytometry cell sorter (BD Biosciences). After FACS, the cells were analysed for their purity by resorting an aliquot of the five purified cell populations generated for MS-based proteomics. The cells were stored as a pellet at −80 °C for later proteomics analysis. For single-cell RNA-seq of CD34-positive cells, single CD34-positive cells were sorted directly into 96-well plates containing 4.4 µl of lysis buffer per well. The lysis buffer contained 0.2% Triton X-100 (Sigma), RNase inhibitor (Takara), oligo-dT$_{30}$VN primer (Sigma) according to Picelli et al.[63] and 2.2 mM dNTP (Invitrogen). The lysed cells were frozen on dry ice cooled ethanol and kept at −80 °C until further processing.

**Isolation followed by culture and sample preparation of human MSCs.** MSCs were isolated using their natural plastic adherence upon culture in vitro[64]. The adopted preparation for MSC has been defined by a consensus position paper of the International Society for Cellular Therapy[65]. MNCs were seeded in a low foetal calf serum (FCS) MSC medium at a density of approximately $1 \times 10^6$ cells per cm² in tissue culture flasks coated with 10 ng ml⁻¹ fibronectin (Sigma) before use. The medium consisted of Dulbecco's modified Eagle's medium with low glucose supplemented with 40% (v/v) MCDB201 (Sigma), 2 mM L-glutamine (Sigma), 100 U ml⁻¹ penicillin/streptomycin (Lonza), 1% (v/v) insulin transferrin selenium (Sigma), 1% (v/v) linoleic acid albumin from bovine serum albumin (Sigma), 10 nM dexamethasone (Sigma), 0.1 mM L-ascorbic acid-2-phosphate (Sigma), homodimer of PDGF subunit B (PDGF-BB) and epidermal growth factor (both 10

ng ml⁻¹; PreproTech, Rocky Hill, NJ, USA), and 2% (v/v) FCS (HyClone). Culture medium was changed twice per week. After initial colony formation after 10–14 days and with 80% confluence, the cells were trypsinized, counted, and reseeded at 10⁴ cells per cm² for further expansion. At passage 2, the cells were scratched off without the use of digesting agents, washed, and cell material stored at −80 °C as a pellet for proteomics analysis. In a series of publications, our group has demonstrated that comparing the MSC preparations from donors of different age groups and harvested in each case after a standardized number of passages will yield reproducible results that demonstrated age-specific changes in epigenetic signatures[17,18,64].

**Sample preparation for proteomics analyses.** The frozen cells were suspended and lysed with lysis buffer by pipetting them up and down at least 30 times. The lysis buffer contained protease inhibitors (Sigma P8340), RapiGest SF surfactant (Waters), and 200 mM 4-(2-hydroxyethyl)piperazine-1-ethanesulfonic acid (HEPES) and was buffered to a pH of 8 with NaOH. Samples were kept at 90 °C for 5 min with subsequent sonication for 20 min. The cell debris was pelleted and the supernatant was further used. Disulphide bonds of the proteins were reduced with dithiothreitol (Biomol) (2 mM) followed by carbamidomethylation of cysteine side chains using iodoacetamide (Merck) (5 mM). The modified proteins were digested first with Lys-C in a 1:100 enzyme to protein ratio (Wako Chemicals) for 3 h at 37 °C with subsequent tryptic digestion in a 1:50 enzyme to protein ratio (trypsin gold, Promega Corporation) at 37 °C overnight. The protein amount was estimated based on the known cell number and determination of the average protein amount per cell for each of the different cell populations.

**Preparation of internal standards for proteomics analyses.** FACS sorted cells from umbilical cord blood and human bone marrow were used to create an internal standard to allow comparison of the individual experiments. For each cell population an individual internal standard was prepared. The lysis and digestion procedure for the internal standard was the same as described above in the section 'Sample preparation for proteomics analysis'. The acid cleavable detergent RapiGest was cleaved after the trypsin digestion with trifluoroacetic acid followed by centrifugation to remove the precipitated lipophilic part of RapiGest. The supernatant containing the peptides was desalted, concentrated, and reconstituted. The individual samples of the same cell population were pooled and aliquoted in

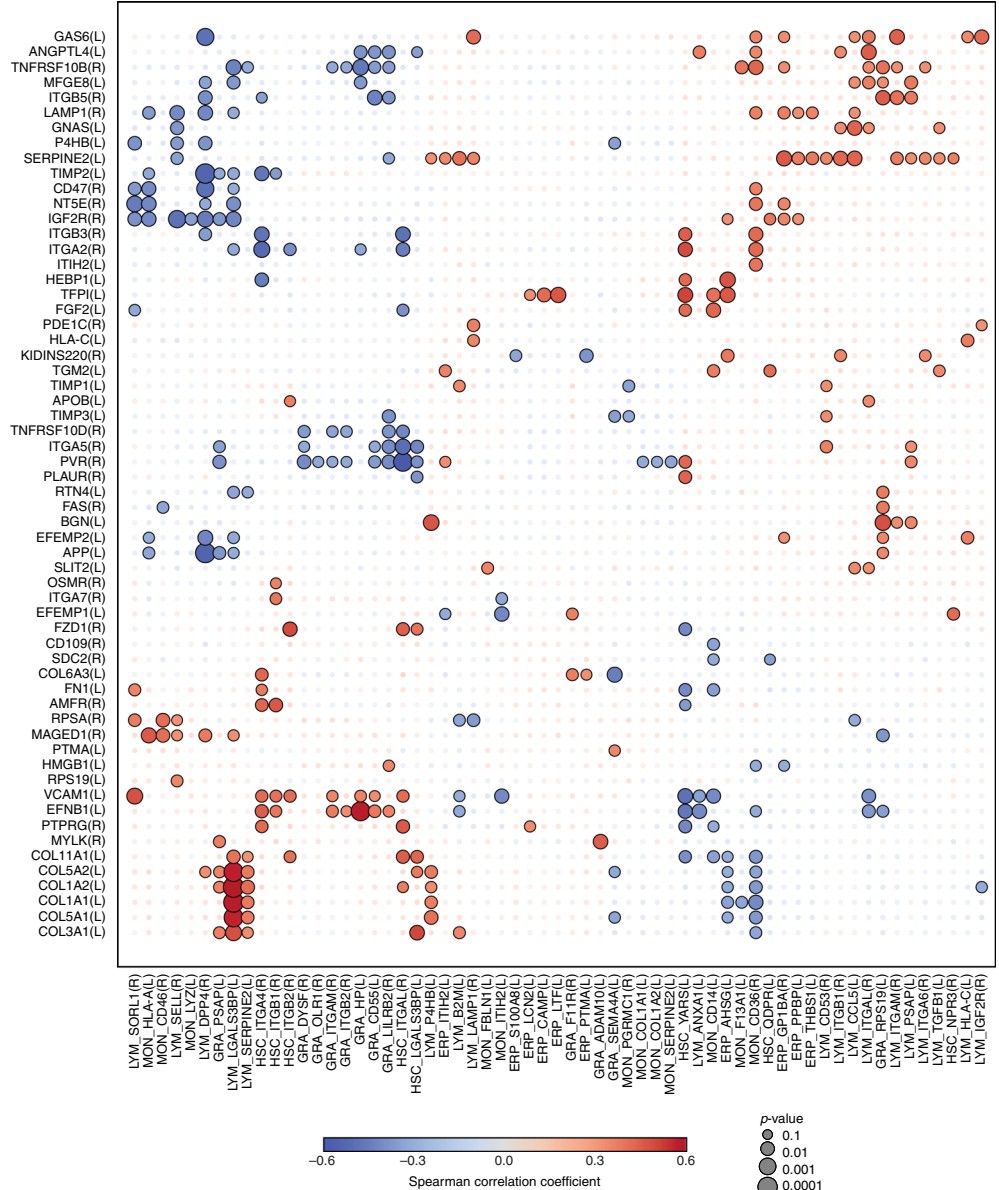

**Fig. 9** Coordinated alterations occur across the different subpopulations upon ageing. Correlation matrix (Spearman) for extracellular protein–ligand receptor pairs at the interface between MSCs and all other cell populations (see Methods). MSC ligands (L) and MSC receptors (R) altered upon ageing (*p* value < 0.1, Spearman's correlation) on the *y*-axis have at least one correlation value with altered receptors (R) or ligands (L) quantified in the other cell populations with a *p* value < 0.1, Spearman's correlation. The colour indicates positive or negative correlation and the size of the circle is reciprocal to the associated *p* value. Correlation coefficients with *p* values ≥ 0.1 are depicted as transparent. Unsupervised clustering using Euclidean distances and the complete clustering method was performed on both axes

amounts equal to the samples. The same batch of aliquots was used for the whole proteomics analysis.

**TMT labelling for protein quantification**. The peptides of five samples of the same cell population and one internal standard with an equivalent amount of sample were modified with TMT (Thermo Fisher Scientific) in order to introduce a label for quantification. The TMT 6-plex has been chosen as it permits measurements of six samples at once and thus reduces the overall run time per sample on the mass spectrometer compared to other labelling strategies that offer less than six channels. After labelling the peptides with TMT, the reaction was quenched with hydroxylamine and the acid cleavable detergent RapiGest was cleaved with trifluoroacetic acid. The lipophilic part of the RapiGest reagent precipitates and was pelleted by centrifugation. The supernatant containing the labelled peptides was desalted on a C18 reversed phase material to remove the buffer, the hydrophilic part of the RapiGest detergent, and other very hydrophilic components.

A small aliquot of the six differentially TMT-labelled samples was mixed and the resulting sample was analysed by liquid chromatography-tandem mass spectrometry (LC-MS/MS). Based on the median protein ratio of each sample over the internal standard, the mixing of the six samples was adjusted as close as possible to a 1:1:1:1:1:1 ratio with the leftover samples.

This mixture of five samples and the internal standard was concentrated under vacuum to remove the majority of organic solvents. Subsequently, the pH was adjusted to above 10 with 25% ammonia and the volume was adjusted to 50 μl. The total sample was separated on an Agilent 1260 infinity high-performance liquid chromatography (HPLC) system equipped with a Waters XBridge C18; 3.5 μm; 1 × 100 mm reversed phase column at a flow rate of 75 μl min$^{-1}$. The buffers were 20 mM ammonium formate at pH 10 and 100% acetonitrile. Ninety fractions were collected and subsequently concentrated under vacuum to remove the majority of the organic solvent. The fractions were desalted and pooled in one step into 18 pooled fractions. Sixteen of these were pooled by taking one early, one middle, and one late eluting fraction and two fractions were created by mixing part of the earliest and latest fractions together. This mixing scheme leads to a higher

orthogonality of the first and second dimension separation and as a consequence uses the available time on the mass spectrometer more efficiently.

**LC-MS/MS analysis of cellular proteomes.** Half of the volume of the 18 pooled fractions was analysed on a Waters nanoAcquity UPLC system directly connected to an Orbitrap Velos Pro (Thermo Fisher Scientific). The utilized columns were a nanoAcquity Symmetry C18, 5 μm, 180 μm × 20 mm trapping column (Waters) and a nanoAcquity BEH C18, 1.7 μm, 75 μm × 200 mm analytical column (Waters). The mobile phases A and B consisted of 0.1% formic acid in water and acetonitrile, respectively. The applied 120 min multi-step gradient ranged from 3 to 85% mobile phase B at a constant flow rate of 300 nl min$^{-1}$. The eluent was directly introduced into the mass spectrometer via a Pico-Tip Emitter 360 μm OD × 20 μm ID; 10 μm tip (New Objective). The applied spray voltage was 2.2 kV and the capillary temperature was set to 300 °C. We measured in positive ion mode. Full scan MS spectra were recorded from 350 to 1500 m z$^{-1}$ in profile mode in the Orbitrap. The resolution in MS mode was set to 30,000. The 10 most intense parent ions were subjected to fragmentation by higher-energy collisional dissociation with the normalized collision energy set to 40. Charge state screening was enabled to prevent analysis of singly charged ions. The resolution for MS/MS was set to 7500.

**Single-cell RNA-sequencing of HPCs.** Sequencing libraries from 192 single CD34$^+$ cells per donor were generated based on the smart-seq2 protocol of Picelli et al.[63] and the tagmentation procedure of Hennig et al.[66] with slight modifications. Single CD34-positive cells were FACS sorted directly into 96-well plates containing 4.4 μl of lysis buffer per well as described above. The lysates were incubated for 3 min at 72 °C and kept on ice while adding reverse transcription (RT) mix. Our RT mix had a final volume of 6.35 μl to have the final reaction volume of 10 μl and in contrast to the smart-seq2 protocol a final concentration of 10 mM MgCl$_2$. Twenty-two cycles were applied for the PCR. The subsequent washing procedure was optimized. Then, 25 μl nuclease-free water and 30 μl of SPRIselect (Beckman Coulter) (1:0.6 ratio) were added and no ethanol wash was performed. After incubation, removal of supernatant, and drying, 13 μl nuclease-free water was applied for elution and 11 μl was taken for a second purification step. Then, 40 μl nuclease-free water and 25 μl of SPRIselect (1:0.5 ratio) were added and after incubation, removal of supernatant, and drying, 13 μl nuclease-free water was applied for elution. Next, 1.25 μl of the supernatant was used for tagmentation[66]. Tn5 was mixed with equal amounts of Tn5ME-A/Tn5MErev and Tn5ME-B/Tn5MErev and incubated at 23 °C for 30 min. Loaded Tn5 and sample were incubated for 55 °C for 3 min in 10 mM Tris-HCl pH 7.5, 10 mM MgCl$_2$, and 25% dimethylformamide. The mixture was cooled to 10 °C and the reaction was stopped with 0.2% SDS for 5 min at room temperature. KAPA HiFi HotStart ReadyMix was used for PCR amplification. Then, 1 μl of each of the 192 samples of one donor was combined and bead purification using 0.8 vol. of SPRIselect was performed including two ethanol washing steps. The elution volume was 50 μl. Sequencing was performed on an Illumina NextSeq 500 with 75 bp single-end reads.

**RNA-sequencing of total HPC populations.** RNA was extracted with trizol (Invitrogen) using a linear acrylamide carrier. RNA was then treated with DNase I (Life Technologies) and purified using Agencourt RNAClean XP beads (Beckman Coulter). RNA quality and concentration were assessed using an RNA 6000 bioanalyzer pico kit (Agilent). Samples with concentrations less than 30 pg μl$^{-1}$ and/or an RNA integrity number less than 6 were excluded from further analysis. A complementary DNA (cDNA) library was produced using the Smart-Seq2 protocol[63]. Sequencing was performed on an Illumina HiSeq4000 with 75 bp paired-end reads with the aim to achieve coverage of 25 million reads per sample.

**Sample preparation for metabolomics.** The metabolites were extracted from cell pellets by cold methanol extraction. Cell pellets of HPCs were suspended in 150 μl −80 °C precooled 80% methanol and kept on dry ice for 20 min. The debris was spun down at 14,000 × g in 5 min at 4 °C and the supernatant was transferred to a new tube. The debris was washed with 50 μl −80 °C precooled 80% methanol for 1 min and the debris was spun down again. The supernatant was pooled with the first supernatant and concentrated in a vacuum centrifuge. This sample is further referred to as metabolite extract.

**Derivatization of phosphorylated sugars.** The metabolic extracts and sugar standards in methanol/water were labelled with 3-amino-9-methylcarbazole (AMC) prior to analysis. In detail, 50 μl of either a sugar mixture consisting of glucose-6 phosphate (Roche), mannose 6-phosphate (Sigma), ribose 5-phosphate (Sigma), ribulose 5-phosphate (Sigma), erythrose 4-phosphate (Sigma), and dihydroxyacetone phosphate (Sigma) or the metabolite extracts were mixed with 50 μl of 25 mM AMC (Enamine, Ukraine), 25 μl 50 mM sodium cyanoborohydride (Sigma), and 10 μl acetic acid. The reaction was kept at 70 °C for 60 min.

**Internal standard preparation for the sugar analysis.** The phosphorylated sugars glucose-6-phosphate (Roche), mannose 6-phosphate (Sigma), ribose 5-phosphate (Sigma), ribulose 5-phosphate (Sigma), erythrose 4-phosphate (Sigma), and dihydroxyacetone phosphate (Sigma) in methanol/water were derivatized as

described above for the metabolite extracts except that a deuterated version of AMC (3-amino-9-methyl-d3-carbazole (AMd3C), Enamine, Ukraine) was used for derivatization. The derivatized sugars were purified as described below and mixed to create a solution with a concentration of 80 fmol μl$^{-1}$ for all sugar phosphates except ribulose 5-phosphate that was 800 fmol μl$^{-1}$.

**Purification of derivatized phosphorylated sugars.** Prior to purification of the AMC-derivatized metabolites from HPCs, 10 μl of the internal standard was added to each sample. Prior to addition to the samples, the internal standard was purified as described in the following: a TiO$_2$ micro-column was packed in a 20 μl GELoader tip (Eppendorf). A small plug of a C8 filter was put at the constricted end of the tip to hold back the TiO$_2$ beads. The 5 μm Titansphere TiO$_2$ beads (GL Science, Japan) were suspended in 0.6% acetic acid in 80% acetonitrile and loaded into the micro-column. The long tip below the C8 plug was cut away to reduce the backpressure of the column. The total HPC sample was loaded onto the micro-column and was washed twice with 30 μl of 0.6% acetic acid in 80% acetonitrile. Subsequently, the sample was washed with 30 μl water and was eluted first with 30 μl 5% ammonia and second with 3 μl 0.6% acetic acid in 80% acetonitrile. The elution was collected directly in a low binding glass vial containing 45 μl 10% trifluoroacetic acid to acidify the eluate. Finally, the sample was concentrated in a vacuum centrifuge.

**LC-MS/MS analysis of phosphorylated sugars of HPCs.** The metabolomics analysis was performed on the same LC-MS/MS equipment as for the proteomics analysis (a Waters nanoAcquity UPLC system directly connected to an Orbitrap Velos Pro, see above) to enable the high sensitivity required to analyse the low metabolite concentrations. The LC gradient and MS method was adopted for AMC-derivatized sugar phosphates. The mobile phases A and B consisted of 0.1% formic acid in water and acetonitrile, respectively. The applied 30 min multi-step gradient increased from 4 to 20% mobile phase B within 3 min followed by an increase to 30% mobile phase B up to 15 min and subsequent washing at 85% and column equilibration at 4% mobile phase B. The flow rate was kept constant at 300 nl min$^{-1}$. We measured in positive ion mode. Full scan MS spectra were recorded from 275 to 700 m z$^{-1}$ in profile mode in the Orbitrap. The resolution in MS mode was set to 30,000. A parent mass list was created that contained the masses of the AMC- and AMd3C-derivatized sugar phosphates. The most intense ions, if no parent masses were found, was enabled and 10 parent ions per cycle were subjected to fragmentation by collision-induced dissociation in the ion trap with the normalized collision energy set to 40. Charge state screening was enabled to prevent analysis of triply and higher charged ions.

**Data analysis of phosphorylated sugars.** The raw files were analysed with Xcalibur 2.2 (Thermo). The ratio between the areas under the peaks of the parent masses in the extracted ion chromatograms and the internal standard was compared with a calibration curve of AMd3C-derivatized sugar phosphate standards. The $R^2$ of the calibration curves was greater than 0.998 for all analyses. Based on standards we analysed during method development we knew that glyceraldehyde 3-phosphate and dihydroxyacetone phosphate coeluted and could not be distinguished. We thus reported the sum of the two. The same was true for mannose 6-phosphate and galactose 6-phosphate.

**Proteomics data analysis.** Supplementary Fig. 13 represents a schematic overview of the data analysis procedures. Two strategies for quantification were employed. Quantification based on TMT for accurate determination of changes upon ageing within the same cell population and a label-free (LF) approach to estimate the protein abundance across proteins and cell populations.

**Data analysis for TMT-based quantification.** MS raw files were first processed with Thermo Proteome Discoverer version 1.4.1.14. The spectra were de-isotoped, deconvoluted, and the mass range from 126 to 131.3 m z$^{-1}$ was excluded prior to database search. The spectra were searched against the Uniprot database including common contaminations (89,016 sequences) using Mascot version 2.5.1 as search engine with trypsin cleavage specificity, one missed cleavage allowed, a precursor mass tolerance of <20 ppm, and a fragment mass tolerance of 0.02 Da. Carbamidomethylation was set as fixed modification and oxidation on methionine was set as variable modification. TMT was implemented in the quantification method as fixed modification. A false discovery rate (FDR) was calculated with Percolator version 2.04. The unprocessed peptide spectrum matches (PSMs) still including the TMT reporter ions and fulfilling the FDR of 1% were exported from Proteome Discoverer and processed by a custom analysis pipeline.

PSMs were filtered to retrieve a high-quality dataset for quantification. Spectra with a search rank greater than one were discarded. For quantification, additional quality filters were applied. Only PSMs with an isolation interference of <30%, the sum of intensities of the TMT channels 127 to 131 (not the internal standard; TMT channel 126) >30,000 and having no missed cleavages were used for quantification. Missing quantification values in the channels 127 to 131 were replaced with the minimum value detected in the corresponding TMT experiment. Given a PSM and its intensities measured in a donor sample (TMT channels 127 to 131), we first calculated its ratio to the internal standard (TMT channel 126). Next, we

normalized each of the ratios by multiplying it with the median ratio determined between the internal standard and the respective donor channel. Note that the median ratios used for normalization are determined using the raw intensities of each channel. Normalized PSM intensity ratios were then used to derive peptide ratios. For peptides with two or more matching PSMs, we considered the median ratio of the three PSMs with highest precursor intensity as final peptide ratio. Peptides were assigned to proteins and proteins were grouped, according to common gene names. These protein groups named after their corresponding gene are called proteins for simplicity throughout the whole publication. Final protein ratios were obtained by taking the median ratio of peptides that are unique for a protein group. Only proteins with at least two unique peptides were considered for quantification. Subsequent bioinformatics analysis has been performed on studentized ratios using the programming language R and Python 2.7.

**Summary of the normalization steps of the TMT data**. The first normalization step was done at the PSM level to correct for slight mixing errors/small differences of input material/under-sampling. For each individual sample (TMT channels 127–131) in each TMT 6-plex experiment, we calculated the median ratio between the internal standard (TMT channel 126) and each sample (TMT channels 127–131). We normalized each individual ratio from all samples by multiplying it with the determined median ratio of the corresponding TMT channel. Note that the median ratios used for normalization are determined using the raw intensities of each channel.

The second normalization was performed on the protein level to correct for differences between the TMT experiments. For each protein the standard deviation and the mean over all TMT experiments of one cell population was used to studentize the values of each ratio. These studentized ratios were used for subsequent bioinformatics analysis, except for the slope calculations for Fig. 3a and Supplementary Fig. 10. For this slope calculation only the mean was used for the ratio normalization. Further details see in the corresponding section 'Statistical analysis of protein expression with age for TMT'.

**Data analysis for LF quantification**. For accurate comparison of the same protein within a cell population, all samples were also analysed with MaxQuant 1.5.3.17[67] to estimate protein abundances. The parameters were: fixed modifications: carbamidomethyl (C), TMT modifications (K, N-term); variable modifications: oxidation (M) and minimum peptide length was seven. Precursor intensities were extracted for each peptide in each separate LC-MS analysis and all associated PSMs were collected. The evidence files were consulted to extract all PSMs with the corresponding intensity value being the summed up extracted ion current (XIC) of all isotopic clusters associated with the identified amino acid sequence. Hence, for each identified peptide in each separate LC-MS analysis, one intensity value was assigned, and all associated PSMs were collected per LC-MS analysis.

For further analysis, we only took into account PSMs that were common to both the TMT quantification approach as described above and the MaxQuant analysis. These spectra were then filtered according to the same criteria as applied for TMT quantification (high confidence, no missed cleavages, search engine rank set to 1, summed intensity of channels 127 to 131 set to more than 30,000, isolation interference set to less than 30%). For each peptide per LC-MS analysis the PSMs that passed latter criteria were collected, and the median ratio was taken as peptide ratio for each LC-MS analysis. If more than three PSMs passed the criteria, the median of the top three most intense spectra was taken.

The total area (MS1 intensity/precursor intensity) of an individual peptide species represents the sum of the internal standard and the five samples that were analysed together. The portion of the total area from each sample can be calculated based on the reporter ion intensities of the six TMT channels. The total area of a given peptide species was split into individual channels using the TMT ratios per peptide per LC-MS analysis. After the splitting of the total area intensity into individual channels, we corrected for potential sampling aberrations by multiplying the area intensities per channel with the median ratio determined between the internal standard and the respective donor channel.

Based on these normalized area intensities per channel, we calculated LF scores for each peptide by dividing through the number of potentially observable unique tryptic peptides per protein (criteria: peptide length 8–25 amino acids, no missed cleavage allowed). In order to retrieve the final LF score per protein, we summed up the LF scores of the corresponding unique peptides per protein. For further data analysis, we only considered proteins that were covered by at least two unique tryptic peptides.

**Summary of the normalization steps of the LF data**. We normalized the LF protein scores to allow comparison between donors across the different cell populations. After log-transforming the data, the median of all LF protein scores was calculated in each TMT-6plex experiment and was subtracted from each LF protein score in each donor. Summarizing the normalized unlogged values for each protein in each cell population and dividing by the number of maximum donors available per cell population gives the LF sum value denoted in the Supplementary Data 2.

**Quality assessment and sample inclusion for proteomics**. The reproducibility was tested by splitting the lysate of one MSC sample into three aliquots prior to digestion. These aliquots were treated like three individual samples. The TMT ratios of these three samples towards the internal standard were employed for calculating Pearson's correlation coefficient. Pearson's correlation coefficient between the first and second (0.9485), second and third (0.9437), and the first and third replicate (0.931) was on average 0.941. The median coefficient of variation of the three ratios per protein was 4.2%.

The median labelling efficiency of all TMT-6plex experiments (each experiment includes five samples and an internal standard) was 98.5%. In order to calculate the labelling efficiency, all experiments were searched with the TMT-label set as variable modification and for each experiment the number of all completely labelled PSMs was divided by the total number of PSMs identified in one experiment (Supplementary Fig. 2a).

The total number of all proteins identified in all 270 samples was 12,158. Exactly 8000 proteins were quantified with TMT of which 7375 were quantified in more than 15% of the donors in at least one cell population. In all, 7585 proteins were quantified by LF quantification of which 6952 were quantified in more than 15% of the donors. The number of proteins identified in each experiment was traced over the whole time of the study and depended on the cell population (Supplementary Fig. 2b).

A principal component analysis (PCA) was performed on the log2-transformed data and the first two principal components (PC1 and PC2) were plotted against each other. Highest density regions (HDR) of 50, 90, 95, and 97% probability were visualized based on Hyndman[68] and samples with >97% probability were defined as outlier and were discarded. We had 15 outliers of which one was a sample analysed in duplicate and was an outlier in both measurements. Thus, 14 samples were discarded and 270 samples (95%) out of the initial 284 samples were used for the study (Supplementary Fig. 2c).

**Statistical analysis of protein expression with age for TMT**. For identifying age-associated proteomic changes within each cell population, we performed Spearman's correlation analysis to detect proteins whose expression changes with age. For each protein with a donor coverage above 15% we calculated Spearman's correlation between the quantified TMT ratios and the respective donor ages to assess its behaviour with respect to age. Positive correlations indicate an increase of the abundance of a protein with advanced age, while negative (or anti-) correlations indicate a decrease of its abundance with age. Proteins with a $p$ value < 0.05 (Spearman's correlation) are considered to be significantly altered upon ageing.

We treated our disproportionate male dataset as sexless and only focussed on age. We thus checked for a possible effect of the gender disparity in the samples by re-analysing the proteomic data after removing all female samples. The criteria for the male-only analysis were identical with the analysis of all samples. The comparison of the results from the male-only vs. all samples is visualized in Supplementary Fig. 9a.

**Hierarchical clustering of proteins based on LF data**. For the hierarchical clustering in Fig. 2b, we used the scipy-python package (python.org) to compute the linkage matrix based on correlation metrics and using the so-called complete clustering[69]. The same clustering has been applied for Supplementary Fig. 5.

**Pathway analysis across cell populations based on LF data**. LF abundances for proteins were leveraged to understand differences between the six cell populations, and not for age differences as these are more reliably analysed by TMT ratios. Filtering criteria for protein inclusion were as strict as for TMT quantification, requiring proteins to be quantified in at least 15% of available donors of a given cell population.

In order to understand whether pathways differ in their abundance or stoichiometry across the different cell populations (Fig. 2c), we applied the following pipeline. We considered proteins quantified in the LF approach (7585 proteins) and mapped those against the Reactome database (http://www.reactome.org/download-data/, February 2017) after filtering the database for pathway sizes of 5–100 proteins. For each cell population, the abundance of a pathway was approximated by the median LF abundance of proteins associated with it. The average of those medians results in the mean abundance of the pathway across the different cell populations (y-axis of Fig. 2c). To estimate the fraction of proteins in a pathway that change in their stoichiometry, we proceeded as follows: for all 270 samples, protein abundances were normalized to the median pathway abundance to avoid any significance stemming from abundance change of the entire pathway. These normalized values, if at least 15% of donors were quantified in an individual cell population, were cross-compared between the cell populations (Wilcoxon test). From the set of $p$ values obtained from those comparisons, the mean $p$ value is calculated. This procedure is iterated across all proteins associated with pathways, and all $p$ values are adjusted thereafter using the Benjamini–Hochberg procedure. To finally obtain the fraction of the pathway that shows a significant alteration between the cell populations, we considered the number of proteins per pathway with an adjusted $p$ value of less than 0.05. Note that this procedure does not take into account cell-type specificity of proteins, and

does not restrict itself on proteins that are expressed throughout all cell populations. The Supplementary Data 3 contains details for further exploration.

**Analysis of protein co-abundance based on TMT data.** Complex annotations are based on Ori et al.[27] who provide a curated list of 279 non-redundant protein complexes based on CORUM and COMPLEAT protein complex sources. Pathway annotations were taken from Reactome as for the pathway enrichment analysis described below (http://www.reactome.org/download-data/, February 2017). Proteins belonging to the same complex were correlated against each other. This was done by calculating Pearson's correlation coefficient for all possible combinations of two members of the same complex. The resulting correlation coefficients from all complexes were plotted in Supplementary Fig. 4. The same procedure was repeated for pathways, with the exception that protein pairs which are already present in a complex were not considered for pathways. The distribution of Pearson's correlation coefficients from complexes was shifted towards the right relative to the distribution of correlation coefficients calculated from all other quantified proteins that were not assigned to complexes or pathways.

**Analysis of pathway changes during ageing based on TMT data.** The Reactome Database (http://www.reactome.org/download-data/, February 2017) was the basis for the analysis of pathways. The displayed pathways of Fig. 3a and Supplementary Fig. 10 were selected based on their size, with the requirement for at least 5 and a maximum of 150 proteins. More than 30% of these needed to be quantified by TMT in at least one cell population and out of the quantified more than 20%, but at least 3 proteins need to be significantly altered upon ageing ($p$ value < 0.05, Spearman's correlation). Thereby, we obtained 28 pathway hits for HPC, 44 for LYM, 3 for GRA, 4 for MON, 2 for ERP, and 221 for MSC. In order to avoid redundancies, we removed pathways whose significantly altered proteins were completely covered in another pathway. If a pathway, however, contained at least one unique significantly altered protein, the pathway was kept. In a scenario of pathways containing exactly the same altered proteins, the largest pathway was reported. In case of equal pathway size, the pathway with a higher hierarchy level was taken. Thereby, we obtained 109 pathways in total that were largely non-overlapping and exhibiting considerable changes upon ageing. In order to decide whether the proteins of a pathway have a general tendency of increasing or decreasing upon ageing, we first calculated the slopes of all proteins based on the linear regression between the donor to internal standard ratio (average normalized per TMT 6-plex experiment) and the age. For the slope calculation normalization by studentization of the protein ratios was avoided and average normalization was applied instead for normalization to avoid artificial high slopes for slightly altered proteins (see section 'Summary of the normalization steps of the TMT data'). The slopes of all significantly altered proteins within a pathway were averaged. Pathways with an average slope of the altered proteins between −0.001 and 0.001 per year of life were reported as having no tendency. A slope of 0.001 translates to an estimated average increase of 4% in protein abundance in a life span from 20 to 60 years (40 years). The results of the 109 pathways were displayed in Supplementary Fig. 10 and listed in Supplementary Data 7. Figure 3a is a selection of these 109 pathways, which is based on taking the five most up- and down-regulated pathways per cell population.

**Cross-cell population correlation analysis based on TMT data.** Extracellular protein–ligand receptor pairs were downloaded from http://fantom.gsc.riken.jp/5/ [56] and overlapped with our TMT-based dataset. Spearman's rank correlation coefficients between ligands and receptors from MSCs and the other cell populations (MSC to HPC, MSC to LYM, MSC to GRA, MSC to MON, MSC to ERP) were calculated. Requirements for inclusion of proteins were that the protein was altered upon ageing ($p$ value < 0.1, Spearman's correlation) in the respective cell population and that the protein profiles (based on studentized TMT protein ratios) of MSC and the respective cell population had at least 85% overlapping donor individuals. Correlation coefficients were visualized in Fig. 9, highlighting correlation results with $p$ values < 0.1 (Spearman's correlation).

**Analysis of the transcriptomics data of HPCs.** Reads were trimmed for Nextera, Smart-seq2 adapter sequences using skewer-v0.1.125[70]. Trimmed read pairs were mapped to human genome hg38.ERCC using HISAT2 version 2.0.0-beta[71]. Uniquely mapped read pairs were counted using featureCounts[72], subread-1.5.0[73], using exons annotated in ENSEMBL annotations, release 75. The subsequent analysis is performed in the programming language R. For the analysis of the raw counts retrieved from RNA-seq experiments, we used the DESeq2 package (version 1.18.1)[74]. We applied a minimal pre-filtering to remove rows that have only 0 or 1 read, as suggested in the DESeq2 manual. To assess the quality of the data, PCA was used after a so-called regularized log transformation (rlog) of the sample, accounting for the library size of each sample. Using HDR plots on the derived principal components similar as described in the section 'Quality assessment and sample inclusion for proteomics', outliers were detected for HPC, GRA, MON, and MSC and removed. For comparison with the corresponding proteomics data, we defined young as ≤30 years of age, and old as ≥50 years of age (similar to Fig. 8b). For each fold change (old/young) a $p$ value (Wald test) was calculated and weighted based on the IHW package[75] that takes into account number of reads as a covariate

for the adjustment of $p$ values. From the output table the log2 fold change was extracted for Supplementary Fig. 9b, which gives an estimate for the effect size (see Supplementary Data 5).

**Single-cell RNA-sequencing data pre-processing.** The single-cell data pre-processing is performed using the programming language R. Raw reads were processed using the recent version of the Salmon pipeline (v0.9.1)[76], with the index derived from transcriptome data from the hg38 build for mapping purposes (http://ftp.ensembl.org/pub/release-87/fasta/homo_sapiens/cdna/Homo_sapiens.GRCh38.cdna.all.fa.gz). The count matrix generated for individual transcripts across cells in each sample was then subjected to further processing using the Bioconductor package tximport[77]. Thereby, the transcript-specific count tables were converted into gene-specific count tables across cells. To filter for qualitative cells, we only retained cells where at least 1000 genes have been found to be expressed at a minimum of 10 reads each, and where the total read count is at least 150,000. That filtering step has been adapted from Velten et al.[78]. Additionally, only genes with at least 10 reads in at least 5 cells were kept for further processing. The resulting count tables were analysed using the Bioconductor package simpleSingleCell (version 1.2.0)[79]. The pipeline was applied with the following steps: (a) additional quality control on cells and filtering due to library size and possible batch effects, and (b) normalization of cell-specific biases using computed size factors. For details on individual samples, the Supplementary Table 3 is to be consulted. The normalized log expression values were further adjusted to the mean expression in each cell.

**Classification of single CD34⁺ cells.** The classification of single CD34$^+$ cells into myeloid- and lymphoid-primed cells is the basis for the analyses in Fig. 6 and Fig. 7. For clustering of single cells, we used Python version 2.7. We first determined whether lymphoid and myeloid markers as delineated in Fig. 5 for the proteomics data were yielding signal in the single-cell RNA-seq dataset. To ensure signal consistency, we excluded markers that did not correlate with the other lymphoid or myeloid markers, respectively ($p$ value < 0.01, Pearson's correlation). Thereby, *ITGA6* had to be removed from lymphoid markers (remaining 12 genes), and *IKZF1*, *ITGAL*, *PRAM1*, and *BCL11A* from myeloid markers (remaining 8 genes). The lymphoid and myeloid markers also had a significant correlation with other known lineage markers, such as *TFRC* (*CD71*) and *CD19* (data not shown). For further analysis *CD71* and *CD19* were also included. To cluster cells into lymphoid/myeloid lineage, cells were required to have at least half of the respective markers stably expressed (>0), and none of either *CD71* or *CD19* (Fig. 6c). That way we could characterize cells in a more conservative manner as being lymphoid-primed or myeloid-primed cells; cells that did not fall into either of those categories were labelled as undefined. When compared to the entire set of cells per donor, we could see that cells defined as lymphoid- or myeloid-primed were significantly different in their marker constellation (average $p$ value (lymphoid) = $9.77 \times 10^{-10}$, average $p$ value (myeloid) = $1.47 \times 10^{-2}$, both Fisher's exact test), whereas this was not the case for the undefined cells (average $p$ value = $7.46 \times 10^{-1}$, Fisher's exact test).

**Lineage- and age-dependent expression of glycolytic enzymes.** Genes derived as age-dependent from the proteomics dataset (Fig. 4) were subsequently analysed on whether they are affected by lineage, or the age of the donor in the single-cell RNA-seq dataset. We correlated genes involved in the glycolysis, tricarboxylic acid (TCA) cycle, and FAO, with lymphoid and myeloid markers, respectively. We found that age-regulated genes ($p$ value < 0.1, Spearman's correlation, Fig. 4) had a stronger disparity between lymphoid and myeloid correlation distribution than genes not found to be age dependent (Fig. 7a) (age dependent: effect size (Cohen) = 1.59, $p$ value = $7.29 \times 10^{-8}$ ($t$-test), age independent: effect size (Cohen) = 0.9, $p$ value = $3.06 \times 10^{-3}$ ($t$-test)). This lineage effect was further examined by calculating the ratio between expression levels in lymphoid vs. myeloid cells for each protein. Age-regulated enzymes of the upper glycolytic pathway ($p$ value < 0.05, Spearman's correlation) were found to be significantly affected by lineage ($p$ value < 0.01, Mann–Whitney $U$-test) across samples, as opposed to enzymes that had no age dependency. Ageing effects on enzyme expression levels were tested in lymphoid and myeloid cells, respectively, to remove lineage effects. The slope calculated from the median expression levels across samples indicated that age-up-regulated glycolytic enzymes are indeed more prone to becoming higher expressed with age, at the single-cell level as well ($p$ value = 0.079, analysis of variance test) (Fig. 7b).

**Code availability.** The code is available as a Supplementary Software file.

## Data availability

The mass spectrometry proteomics data have been deposited at the ProteomeXchange Consortium via the PRIDE partner repository with the dataset identifier PXD007048. Raw data for both the single-cell RNA-seq and bulk RNA-seq experiments have been deposited in the Gene Expression Omnibus (GEO), database under accession code GSE115353. The authors declare that all data supporting the findings of this study are available within the article and its supplementary information files or from the corresponding author upon reasonable request.

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

## Acknowledgements

We are grateful to the EMBL Proteomics Core Facility, the EMBL Genomics Core Facility, the EMBL Chemical Biology Core Facility, the Wellcome Trust Centre for Human Genetics Bioinformatics Core, Martin Beck's group, Linda Manta, Noorie Karimbocus, and Luis Ferrandez Peral for expert help. We also thank Christophe Lancrin and other members of A.-C.G., A.D.H., P.A. and P.B. groups for continuous discussions and support. We thank all groups and group leaders in the EMBL Structural and Computational Biology Unit for inspiring discussions and creating a stimulating and vibrant environment. The work is funded mainly by the EU-FP7 grant (306240) SysTemAge, and partly by the EMBL, as well as by the SFB873 (Project B7 for A.D.H.), Deutsche Forschungs-Gemeinsch. A.P. and J.B. acknowledge support from Bloodwise (UK; grant 13042).

## Author contributions

A.D.H., A.-C.G., P.A., and T.L. designed the project. A.D.H., A.-C.G., M.L.H., P.A., P.H., J.B.Z., and P.B. supervised and administered the project and wrote the manuscript with the help of all the authors. P.H. and L.P.-B. processed the bone marrow aspirates and P. H., X.D., and L.P.-B. in vitro expanded MSCs. V.E. sorted the cells by FACS. X.D. analysed the purity of the sorted cell populations. M.L.H. established the proteomics pipeline and M.L.H. and F.Y. performed the proteomics analysis. V.S., J.B., and A.P. performed the transcriptomics studies. M.L.H. and M.C.L. performed the single-cell transcriptomics analyses. M.L.H. developed the metabolomics method. M.L.H. and X.D. performed the metabolomics analyses. M.L.H., S.J., and N.R. designed the data analysis pipeline. S.J. and N.R. computationally analysed the TMT and the LF data. N.R. performed the different data integration. B.L. performed preliminary data analysis. All authors have read and approved the manuscript.

## Additional information

**Competing interests:** The authors declare no competing interests.

