## [Peer Review File · Nature Communications]

Reviewers' Comments:

Reviewer #1:

Remarks to the Author:

Synopsis:

In the current study, Hennrich et al. present a proteomic map of different hematopoietic cell populations and bone marrow stromal cells, and characterize aging-related changes. They find that in hematopoietic stem and progenitor cells (HPCs) the levels of several glycolytic enzymes increase with age, and further confirm known age-induced changes such as increased levels of myeloid differentiation-related proteins. In addition, they find that aging also causes changes in the stromal cells, particularly in proteins that play a role in HPC homing and differentiation. The dataset presented is unique, and will be of value to the field. Although the analysis performed is limited and mostly descriptive in nature, this study will certainly trigger new investigations into the functional relevance of the observed changes. There are key issues though that should be addressed prior to publication.

Major comments:

- MSCs were isolated through plastic adherence (long term culture). Stromal cells are known to undergo dramatic changes in culture and some aging-related differences may disappear while others appear without necessary correlation in vivo. The authors should discuss the rationale behind choosing this approach versus using freshly isolated stromal cells. They should also add a strong caveat to the discussion and abstract as to the effects the in vitro culture may have had on the proteomics results.

- Around 7000 proteins were measured in this study, but it is estimated that around 20,000 proteins exist in the human body. Although it can be expected that not all existing human proteins are present in the hematopoietic system, the authors should discuss how this incomplete coverage may impact the outcome of their study. For example, it seems that none of the HOX transcription factors was measured, while several members of this family are well-known regulators of hematopoiesis. Does a bias exist in the used methodology towards certain classes of proteins (enzymes, transcription factors, cytokines etc.) or cellular compartments (cytoplasm, nucleus, membrane, mitochondria etc.)? This should be addressed.

- When comparing the basic proteome of the different cell populations the authors discuss only glycolysis (Fig. 2). It makes sense that glycolysis is highlighted given the changes in this pathway during aging (Fig. 4), but the authors should also include a detailed analysis of differences in other pathways (and/or specific proteins) that exist between the different cell populations (cfr. Suppl. Fig. 6).

- The changes in protein levels that are observed with aging may relate to changes in the levels of each protein per cell, but could equally be attributed to changes in the cellular composition of each fraction, especially since the markers/properties that were used to separate the cell populations are not very specific. For example, the CD34+ fraction is known to encompass hematopoietic stem cells, multipotent progenitors and different committed progenitor populations. The increase in glycolytic enzyme levels could therefore represent an increase of these enzymes on a per-cell basis, but could equally be due to the expansion of one of these subpopulations that is more glycolytic than the others. This should be addressed, either by showing that the cellular composition of the fractions does not change with aging, or, preferentially, by quantifying the levels of different glycolytic enzymes using a technique that has the ability to simultaneously distinguish subpopulations such as flow cytometry.

Minor comments:

- Abstract: The authors write: "For each, the abundance of a large fraction of the ~12,000 proteins identified was assessed over a time span of 40 years." This sentence is somewhat misleading and should be changed.

- Results, page 4, line 153-154: The authors claim that they are the first to show Nestin expression in human MSCs, however, this has been shown before: Pinho, S. et al. J. Exp. Med. 2013, 210, 1351-1367.

- Results, page 6, line 217-226 & page 9, line 345-348: The authors discuss changes in the levels of specific proteins, such as DNMT1, IRF8 and TGFbeta1. While the data are included in the manuscript, you have to dig through the supplementary tables to find them. Readability would significantly improve if these results are also presented in a graph format in the main or supplementary figures.

- Discussion, page 10, line 368: The authors report "an enhanced metabolic and anabolic activity of older HPCs". However, no metabolic activity assays are performed, and the authors only show increases in enzyme levels. While both may correlate, this is not always the case. The authors should therefore change the wording, or, ideally, perform enzyme activity assays in young vs old HPCs.

- Figure 3: which criteria were used for selecting the subset of pathways shown in Fig. 3 from the complete list in Suppl. Fig. 10?

- Figure 4: the light red arrow is noted as "downregulated upon aging $0.05 < p < 0.1$ ", this probably should be "upregulated upon aging $0.05 < p < 0.1$ ".

Reviewer #2:

Remarks to the Author:

Henrich ML et al. undertake a well organized, extensive description of age-related changes in the proteome of 6 human bone marrow main cell populations (HPC, LYM, MON, GRA, ERP and MSC). This is the first analysis of this type to my knowledge. The authors have also employed a wide range of analytic tools. The sample preparation is controlled at all the critical steps and low quality data are removed from the analysis. Detailed description of the methods and quality check shows a good attention to detail and ensures high quality results.

The main findings of the paper recapitulate in many regards the results known previously from murine studies, for example a myeloid-bias or a decrease of homing receptor-adhesive interactions with ageing. As a minor comment for accuracy in P.4 L.153, NESTIN expression has been previously reported in human HSC niche-forming MSCs (doi: 10.1016/j.celrep.2013.03.041). Less known are the highlighted changes in central carbon metabolism in aged hematopoietic progenitors. Importantly, not only hematopoietic cell populations, but also stromal cell populations have been analysed, which allowed an interesting approach of combining the obtained data into a correlation matrix (Supp. Fig 12.). The aim of this approach was to highlight which particular extracellular interactions between MSCs and a hematopoietic population are changing during ageing. This is an interesting analysis and it would be good to expand and, if possible, move the figure to the main text.

It would be interesting to know if the authors have observed differences when segregating the samples depending on gender, as there is a documented bias for different age-related hematological conditions. In the future it would be good to expand to older donors (over the age of 60), if possible.

Altogether, the study provides a comprehensive analysis of proteome changes in aged bone marrow cell populations and will serve as a resource reference within this research area.

Reviewer #3:

Remarks to the Author:

This manuscript uses quantitative proteomics approaches to study changes in human bone marrow cells taken from healthy human subjects where 45 male and 14 female subjects donated bone marrow. In the study, the authors separated out specific cells and studied a total of six different cell populations. The authors used both a label free quantitative proteomics approach and the tandem mass tag approach to attempt characterize differences in cell type specific changes (using a label free approach) and age associated changes (using the TMT approach). Pathway and complex analysis using CORUM, COMPLEAT, and Reactome were used to determine the changes in protein expression in ageing, and the major conclusion appears to be that central carbon metabolism pathways are altered and rerouted as HPCs became older. This is certainly an impressive collection of human samples and a large number of proteins identified, but there are a number of issues that the authors must consider prior to publishing this work in any journal.

A major first issue is with the data analysis itself. It is unclear from the manuscript and the methods section how the samples were processed and analyzed. To begin, these are essentially clinical samples from a disproportionately male sample set. A major question is how comparisons were made from a sex as a biological variable standpoint. Were the samples treated essentially as 'sexless' and only a focus on the age was considered? Or was some other approach used. One issue with a skewed sample dataset like this is are the results skewed given the disproportionate numbers of males include in the study? Are they seeing effects that can be explained by differential male and female protein expression?

The next major issue with the data analysis is with respect to the proteomics data processing. To begin, the authors need much better technical explanation for how TMT and LFQ was used individually and combined. In general the methods sections is poor and lacking details. For example, the authors' state that they used a cell-population specific internal standard was added to each pool, but no details on what this internal standard was and how it was prepared is provided. Furthermore, it is unclear what statistical tools were used to determine which proteins were statistically significantly different between samples and what the criteria for inclusion was? It is unclear exactly what is being presented? Is all data only those with a p value of less than 0.01? 0.05? The methods section in the supplement, for example, states "Subsequent bioinformatics analysis has been performed on studentized ratios using R statistical computing software and Python 2.7." Only details are provided for TMT, no statistical info provided for LFQ. The manuscript needs detailed data processing schema, citations for pipelines, a detailed method analysis, and a clear justification for inclusion of protein expression changes for subsequent analysis. The manuscript does provide statistical information for pathway enrichment and cross-cell population correlation in methods in the supplement but it is unclear how they determined exactly what proteins were significantly different using TMT and/or LFQ and how this was determined.

Next, the normalization process needs to be carefully explained. If there are an uneven number of samples from a given cell type at a certain age, which appears to be the case from supplemental table 3, for example, how do the authors account for changes that may appear due to undersampling since they have less material to analyze? Given the nature of proteomics if the same number of cells and/or total protein starting material is not used, changes from one cell type to another could possibly be explained by having more total material for analysis from one sample to another. The authors need to clearly explain how they corrected for this potential confounding issue.

The last major issue for the manuscript is assuming all of the above issues can be clarified, the manuscript contains proteomics data alone and simply compares the results to prior literature. It is unclear what the major new knowledge is here given the extensive literature on ageing. The mitochondria play an important role in ageing (Finkel T, Nature Medicine 2015), for example, but little description of the mitochondria changes, if any, are described here. Also, there is no

validation of biological insights using alternative approaches. Comparing their results to existing transcriptomics and epigenetic data (Refs 17 and 18) could be insightful. Carrying out assays to demonstrate changes in glycolysis of cells would be valuable. Even if the protein levels of central carbon metabolism are altered, no data is presented to determine what effect this has on central carbon metabolism itself. It is assumed that carbon metabolites would be altered, but not evidence is provided. A detailed follow up on a metabolic pathway would be an excellent addition to this body of work.

We are grateful for the constructive comments of the reviewers. In the following, we will address all the issues raised point by point:

Reviewer #1

Synopsis:

In the current study, Hennrich et al. present a proteomic map of different hematopoietic cell populations and bone marrow stromal cells, and characterize aging-related changes. They find that in hematopoietic stem and progenitor cells (HPCs) the levels of several glycolytic enzymes increase with age, and further confirm known age-induced changes such as increased levels of myeloid differentiation-related proteins. In addition, they find that aging also causes changes in the stromal cells, particularly in proteins that play a role in HPC homing and differentiation.

The dataset presented is unique, and will be of value to the field. Although the analysis performed is limited and mostly descriptive in nature, this study will certainly trigger new investigations into the functional relevance of the observed changes. There are key issues though that should be addressed prior to publication.

We appreciate the comment of Reviewer #1 that "...the dataset presented is unique and will be of value to the field...."

Major comments:

1) MSCs were isolated through plastic adherence (long term culture). Stromal cells are known to undergo dramatic changes in culture and some aging-related differences may disappear while others appear without necessary correlation in vivo. The authors should discuss the rationale behind choosing this approach versus using freshly isolated stromal cells. They should also add a strong caveat to the discussion and abstract as to the effects the in vitro culture may have had on the proteomics results.

We agree with Reviewer #1 that it would have been preferable, if freshly isolated stromal cells (MSC) could be used for the proteomics analysis, similar to the procedures for the other populations studied. The rationale behind culturing the MSCs is: Present day technology does not permit isolation of an adequate number of MSC for proteomics (or for most other -omics studies) because (a) a specific marker for MSC has not yet been defined, and (b) this population is extremely rare. The preparation procedure for MSC that we adopted has been defined in a position paper of the *International Society for Cellular Therapy*¹.

We also agree with the reviewer that stromal cells undergo replicative senescence upon culture and after each "passage"². In a series of publications, our group has demonstrated that comparing the MSC preparations from donors of different age groups and harvested in each case after a standardized number of passages will yield reproducible results demonstrating age-specific changes in epigenetic signatures²⁻⁶. We have now clarified this point in the Methods section (*Isolation, culture and sample preparation of human mesenchymal stem/stromal cells (MSCs)*) and added a caveat on page 4 (last paragraph) of the main text.

2) Around 7000 proteins were measured in this study, but it is estimated that around 20,000 proteins exist in the human body. Although it can be expected that not all existing human proteins are present in the hematopoietic system, the authors should discuss how this incomplete coverage may impact the outcome of their study. For example, it seems that none of the HOX transcription factors was measured, while several members of this family are well-known regulators of hematopoiesis. Does a bias exist in the used methodology towards certain classes of proteins (enzymes, transcription factors, cytokines etc.) or cellular compartments (cytoplasm, nucleus, membrane, mitochondria etc.)? This should be addressed.

Reviewer #1 raised the issue how the incomplete coverage of proteins might impact the outcome of the study and whether a bias exists in the methodology towards certain classes of proteins or cell compartments.

Overall, 12,000 proteins were identified and more than 7,000 could be quantified with either TMT-based or label free methods. In a new Supplementary Fig. 3, we have summarized the numbers and percentages of the different categories of proteins covered according to: (a) their corresponding mRNA expression levels in the bone marrow tissue⁷, (b) their functional classes, and (c) their cellular compartments. Whereas our technology was not powered to capture the least abundant proteins (e.g. proteins encoded by mRNAs with ≤ 1 FPKM (fragments per kilobase of transcript per million mapped reads) such as transcription factors), our analyses have covered to a large extent (>50%) the main functional and compartmental protein categories. These new figures have provided evidence that the results are not overly biased towards certain classes of proteins or cell compartments. This analysis is now described on page 3 (third paragraph) of the revised manuscript.

3) When comparing the basic proteome of the different cell populations the authors discuss only glycolysis (Fig. 2). It makes sense that glycolysis is highlighted given the changes in this pathway during aging (Fig. 4), but the authors should also include a detailed analysis of differences in other pathways (and/or specific proteins) that exist between the different cell populations (cfr. Suppl. Fig. 6).

We agree with Reviewer #1 that while describing the basic proteomes, we should focus not only on the most prominent changes in pathways associated with ageing, but should include a detailed analysis of cell specific differences in other pathways, as well. In the revised manuscript, we added a “bird’s eye view” that highlights the extent to which the different pathways are changed in their stoichiometry in different cell populations. This is now described on page 5 (first paragraph), in a new panel (panel c) in Figure 2 of the main text, and in a Supplementary Table 4. The stoichiometry of many metabolic pathways are specific to a cell population (Fisher Exact test, p -value= 4.2×10^{-3}). One of the most prominent examples was glucose metabolism (highlighted with an arrow), which justified our current focus. We feel strongly that a detailed discussion of other differences is far beyond the scope of the present manuscript and should be the topics of further investigations and reports.

4) The changes in protein levels that are observed with aging may relate to changes in the levels of each protein per cell, but could equally be attributed to changes in the cellular composition of each fraction, especially since the markers/properties that were used to separate the cell populations are not very specific. For example, the CD34+ fraction is known to encompass hematopoietic stem cells, multipotent progenitors and different committed progenitor populations. The increase in glycolytic enzyme levels could therefore represent an increase of these enzymes on a per-cell basis, but could equally be due to the expansion of one of these subpopulations that is more glycolytic than the others. This should be addressed, either by showing that the cellular composition of the fractions does not change with aging, or, preferentially, by quantifying the levels of different glycolytic enzymes using a technique that has the ability to simultaneously distinguish subpopulations such as flow cytometry.

We agree with Reviewer #1 that the increase in glycolytic enzyme levels could be due to the expansion of one of these subpopulations, which is more glycolytic than the others, rather than being due to an increase of these enzymes on a per-cell basis. Other authors have demonstrated that even though the total number of CD34+ cells remained stable with age, there was an increase in the number of CD34+/CD38- cells, (statistically significant in one study⁸ and not significant in another⁹). Whereas early and committed myeloid progenitors persisted at the same level, the frequency of committed B-lymphoid progenitors decreased with age⁸. However, this was again not significant in the report by Pang et al.⁹. In summary, the relative expansion of individual subpopulations with age was at most at a minor scale, whereas the lineage skewing of the CD34+ cells towards myeloid development is significant as demonstrated in our study, as well as in the report of Pang et al.⁹.

Reviewer #1 addressed the question, if the increase in glycolytic enzymes in old HPCs could be due to the expansion of one of the subpopulations of CD34+ cells. For this purpose, we have analyzed the transcriptome of 519 single-sorted HPCs (CD34+ cells) originating from young (n=2) and old (n=2) human subjects. Based on abundance levels of the mRNA markers associated with lymphoid or myeloid differentiation as described in Figure 5c (and that we could consistently measure in single cells), we categorized each of the 519 HPCs. The data are shown in a new Supplementary Fig. 12a-c. These results have demonstrated that the mRNA abundance of glycolytic enzymes that increased upon ageing were expressed at significantly higher levels in myeloid-primed than in lymphoid-primed

HPCs, whereas age-unaffected enzymes have similar mRNA levels in both subsets of HPCs (Supplementary Fig. 12d,e and Fig. 5d,e). Remarkably, the age-dependent increase in glycolytic enzymes was most prominent in the myeloid-primed subset of HPCs. Thus, the lineage skewing of the HPCs towards myeloid differentiation observed upon ageing may account, at least in part, for the increase in abundance of glycolytic enzymes.

These new data are now incorporated in Figure 5d,e, in Supplementary Figure 12a-e, and are discussed on page 9 (third paragraph) of the main text.

Minor comments:

5) *Abstract: The authors write: "For each, the abundance of a large fraction of the ~12,000 proteins identified was assessed over a time span of 40 years." This sentence is somewhat misleading and should be changed.*

This is revised and has been changed into: *"For each, the abundance of a large fraction of the ~12,000 proteins identified was assessed in a cohort of 59 human subjects from different ages."*

6) Results, page 4, line 153-154: The authors claim that they are the first to show Nestin expression in human MSCs, however, this has been shown before: Pinho, S. et al. J. Exp. Med. 2013, 210, 1351-1367.

In human bone marrow, albeit nestin+ cells were described in fetal marrow of some 20 gestational weeks in the mentioned paper, the proof of nestin+ cells among adult MSCs has been controversial. We have modified the text on page 4 (last paragraph) and cited the reference as suggested by Reviewer #1.

"Abundant and specific expression of nestin (NES) was found in human MSCs in the present study. The presence of nestin-positive MSCs has been reported to characterize a perivascular bone marrow niche, which supports HPC maintenance and homing in murine models^{10, 11} and in human bone marrow^{12, 13}. Our study has confirmed the existence of nestin-positive MSCs in adult marrow."

7) *Results, page 6, line 217-226 & page 9, line 345-348: The authors discuss changes in the levels of specific proteins, such as DNMT1, IRF8 and TGFbeta1. While the data are included in the manuscript, you have to dig through the supplementary tables to find them. Readability would significantly improve if these results are also presented in a graph format in the main or supplementary figures.*

As suggested by Reviewer #1, we have added a panel b to the Supplementary Fig. 10 that shows the changes in DNMT1 and IRF8. The data concerning TGFbeta1 are now included in a new pane of panel b in Fig. 6.

8) *Discussion, page 10, line 368: The authors report "an enhanced metabolic and anabolic activity of older HPCs". However, no metabolic activity assays are performed, and the authors only show increases in enzyme levels. While both may correlate, this is not always the case. The authors should therefore change the wording, or, ideally, perform enzyme activity assays in young vs old HPCs.*

To address this point, we performed metabolomics analyses of HPCs (n=10, age: 21-69 year old). This required the development of a new method - based on the derivatization of sugar-phosphates with 3-amino-9-methylcarbazole – which was sensitive enough to detect metabolites in only ~100,000 cells with the limit of detection being in the low femtomolar range (the method is described in detail on page 5-6 and page 12 in the Methods section). The following metabolites were measured: glucose 6-phosphate, ribose 5-phosphate, ribulose 5-phosphate, erythrose 4-phosphate, dihydroxyacetone-phosphate / glyceraldehyde 3-phosphate, and mannose 6-phosphate / galactose 6-phosphate. The data showed a tendency for ribose 5-phosphate and ribulose 5-phosphate (in the pentose phosphate pathway) to accumulate in aged HPCs, whereas the other analyzed metabolites showed no age correlation. This is compatible with the notion that the age-associated increase in abundance of enzymes, catalyzing the rate-limiting steps of the upper part of glycolysis, has led to increased fluxes through the pentose phosphate and nucleic acid

synthesis pathways. The results are added to Fig. 4 and are described on page 7 ('Results') and page 10-11 ('Discussion') in the main text.

9) *Figure 3: which criteria were used for selecting the subset of pathways shown in Fig. 3 from the complete list in Suppl. Fig. 10?*

As suggested by Reviewer #1, we have added the criteria for selecting the subset of pathways in the legend of Fig. 3.

10) *Figure 4: the light red arrow is noted as "downregulated upon aging $0.05 < p < 0.1$ ", this probably should be "upregulated upon aging $0.05 < p < 0.1$ ".*

We apologize for this mistake and thank the reviewer for spotting it. This is corrected in the revised manuscript.

Reviewer #2

Hennrich ML et al. undertake a well-organized, extensive description of age-related changes in the proteome of 6 human bone marrow main cell populations (HPC, LYM, MON, GRA, ERP and MSC). This is the first analysis of this type to my knowledge. The authors have also employed a wide range of analytic tools. The sample preparation is controlled at all the critical steps and low quality data are removed from the analysis. Detailed description of the methods and quality check shows a good attention to detail and ensures high quality results.

We deeply appreciate the comments of Reviewer #2 that this study is a well-organized, comprehensive proteomics analysis of age-related alterations in human marrow cell populations, and this is the first analysis of this type with high quality results.

1) The main findings of the paper recapitulate in many regards the results known previously from murine studies, for example a myeloid-bias or a decrease of homing receptor-adhesive interactions with ageing. As a minor comment for accuracy in P.4 L.153, NESTIN expression has been previously reported in human HSC niche-forming MSCs (doi:10.1016/j.celrep.2013.03.041). Less known are the highlighted changes in central carbon metabolism in aged hematopoietic progenitors. Importantly, not only hematopoietic cell populations, but also stromal cell populations have been analysed, which allowed an interesting approach of combining the obtained data into a correlation matrix (Supp. Fig 12.). The aim of this approach was to highlight which particular extracellular interactions between MSCs and a hematopoietic population are changing during ageing. This is an interesting analysis and it would be good to expand and, if possible, move the figure to the main text.

We agree with Reviewer 2 that the major novelties are (a) the alterations in central carbon metabolism in ageing HPC; and (b) the complementary decrease in levels of adhesive molecules and respective ligands in the HPC and MSC, whereas other findings like lineage skewing have confirmed observations that were thus far predominantly described in murine models of ageing. We also appreciate the comments of Reviewer #2 that the simultaneous and comprehensive analyses of haematopoietic cell populations and stromal cell populations have permitted an interesting approach of combining the obtained data into a correlation matrix. In the revised manuscript, we have significantly expanded the analysis of the major novelties:

(i) The alterations in the abundance of enzymes in central carbon metabolism are now supported by additional transcriptomics data, which confirmed the increase in abundance of the respective mRNA for these enzymes in elderly HPCs (new Supplementary Fig. 9b,c).

(ii) We performed transcriptomics analyses of single-sorted HPCs. These have demonstrated that the age-associated increase in abundance of the glycolytic enzymes was most prominent in the myeloid-primed HPCs (Fig. 5d,e Supplementary Fig. 12a-e; see also page 9 (third paragraph) of the main text).

(iii) We performed metabolomics analyses of HPCs (n=10, age: 21-69 year old). This required the development of a new method - based on the derivatization of sugar-phosphates with 3-amino-9-methylcarbazole – which was sensitive enough to detect metabolites in only ~100,000 cells with the limit of detection being in the low femtomolar range (the method is described in detail on page 5-6 and page 12 in the Methods section). The following metabolites were measured: glucose 6-phosphate, ribose 5-phosphate, ribulose 5-phosphate, erythrose 4-phosphate, dihydroxyacetone-phosphate / glyceraldehyde 3-phosphate, and mannose 6-phosphate / galactose 6-phosphate. The data showed a tendency for ribose 5-phosphate and ribulose 5-phosphate (in the pentose phosphate pathway) to accumulate in aged HPCs, whereas the other analyzed metabolites showed no age correlation. This is compatible with the notion that the age-associated increase in abundance of enzymes catalyzing the rate-limiting steps of the upper part of glycolysis has led to increased fluxes through the pentose phosphate and nucleic acid synthesis pathways. The results are added to Fig. 4 and are described on page 7 ('Results') and page 10-11 ('Discussion') in the main text.

(iv) As suggested by Reviewer #2, we have moved the Supplementary Fig. 12 to the main text. It is now Fig. 6c. We added the suggested reference Isern et al.¹³ (doi:10.1016/j.celrep.2013.03.041) about nestin+ MSCs described in human bone marrow to the main text (page 4, last paragraph).

2) It would be interesting to know if the authors have observed differences when segregating the samples depending on gender, as there is a documented bias for different age-related hematological conditions. In the future it would be good to expand to older donors (over the age of 60), if possible.

We agree with Reviewer #2 that we should expand the recruitment of elderly human subjects over the age of 60 in the future. Indeed, this is being performed. Our present collection reported in the study with 14 female out of a total of 59 human subjects is not powered to allow a segregated analysis to discern gender differences. We have re-analysed the TMT-based quantification by excluding the 14 female subjects and have found similar significant results. For example, we have identified the same alterations with a correlation of up to 0.9 (for glycolytic enzymes), indicating that these changes in the glycolytic pathway are not confounded by gender differences. This analysis is now described on page 6 (first paragraph) of the main text and in a new Supplementary Fig. 9a.

We would like to express our thanks to Reviewer #2 for his comment that “...altogether, the study provides a comprehensive analysis of proteome changes in aged bone marrow cell populations...”.

Reviewer #3

This manuscript uses quantitative proteomics approaches to study changes in human bone marrow cells taken from healthy human subjects where 45 male and 14 female subjects donated bone marrow. In the study, the authors separated out specific cells and studied a total of six different cell populations. The authors used both a label free quantitative proteomics approach and the tandem mass tag approach to attempt characterize differences in cell type specific changes (using a label free approach) and age associated changes (using the TMT approach). Pathway and complex analysis using CORUM, COMPLEAT, and Reactome were used to determine the changes in protein expression in ageing, and the major conclusion appears to be that central carbon metabolism pathways are altered and rerouted as HPCs became older. This is certainly an impressive collection of human samples and a large number of proteins identified, but there are a number of issues that the authors must consider prior to publishing this work in any journal.

We appreciate the reviewer's comment that "...This is certainly an impressive collection of human samples and a large number of proteins identified...."

1) *A major first issue is with the data analysis itself. It is unclear from the manuscript and the methods section how the samples were processed and analyzed. To begin, these are essentially clinical samples from a disproportionately male sample set. A major question is how comparisons were made from a sex as a biological variable standpoint. Were the samples treated essentially as 'sexless' and only a focus on the age was considered? Or was some other approach used. One issue with a skewed sample dataset like this is are the results skewed given the disproportionate numbers of males include in the study? Are they seeing effects that can be explained by differential male and female protein expression?*

Our original overarching goal was to define the systems biology of ageing by comprehensive -omics especially proteomics studies. We focused meticulously on quality of the human samples and the quality of the proteomics analysis. Reviewer #3 is right that "...the samples were treated essentially as 'sexless' and only a focus on the age was considered..." The human subjects were randomly recruited with focus on distribution according to age groups. We clarify this point in the Methods section (second paragraph of the sub-section 'Analysis of protein expression with age and statistical significance calculation (TMT)'): "...We treated our disproportionate male dataset as sexless and only focused on age..."

In addition, we have re-analyzed the TMT-based quantification by excluding the 14 female subjects and have found similar significant results as the original analysis of the whole study population (see Supplementary Fig. 9a). Thus, the novel effects observed cannot be explained by differential male and female protein expression in these specific pathways. This new analysis is described on page 6 (first paragraph) of the main text, and in the second paragraph of the sub-section 'Analysis of protein expression with age and statistical significance calculation (TMT)' of the Methods section.

2) *The next major issue with the data analysis is with respect to the proteomics data processing. To begin, the authors need much better technical explanation for how TMT and LFQ was used individually and combined. In general the methods sections is poor and lacking details. For example, the authors' state that they used a cell-population specific internal standard was added to each pool, but no details on what this internal standard was and how it was prepared is provided. Furthermore, it is unclear what statistical tools were used to determine which proteins were statistically significantly different between samples and what the criteria for inclusion was? It is unclear exactly what is being presented? Is all data only those with a p value of less than 0.01? 0.05? The methods section in the supplement, for example, states "Subsequent bioinformatics analysis has been performed on studentized ratios using R statistical computing software and Python 2.7." Only details are provided for TMT, no statistical info provided for LFQ.*

The manuscript needs detailed data processing schema, citations for pipelines, a detailed method analysis, and a clear justification for inclusion of protein expression changes for subsequent analysis. The manuscript does provide statistical information for pathway enrichment and cross-cell population correlation in methods in the supplement

but it is unclear how they determined exactly what proteins were significantly different using TMT and/or LFQ and how this was determined.

In response to the concern of Reviewer #3 that our Methods section was slightly confusing and lacking technical details, we have extensively rewritten, reorganized and expanded the Methods section especially the part on informatics and statistics analyses. As suggested by Reviewer #3, we included a comprehensible overview of all applied methods in the form of a detailed data processing scheme (see Supplementary Fig. 13) in the revised manuscript. The scheme has four sub-sections, that describe the proteomics pipeline (panels a,b), the new single-cell RNA-seq pipeline (panel c) and the transcriptomics analysis proceedings (panel d). To facilitate the navigation through the methods section, we also organized it better and added clearly labeled subsections (e.g. all statistical methods are described in independent sections, see below). We also revised the section '*Quality assessment and sample inclusion in the proteomics analysis*' (page 9). A legend to each of the Supplementary Tables has been included (previously this was embedded in the Supplementary material). Herein, every column and encoded value are explained.

We also like to address the following points:

(i) We clarified the way TMT- and LF-quantifications were used. We developed a new method to extract LF values from TMT-labelled pooled samples (each LF value originates from 6 samples: 5 donors and 1 internal standard). This is described in detail in the revised section '*Data analysis for label-free (LF) quantification*' (page 8). Importantly, after this step, data acquired with the TMT- and LF- quantification were kept and handled separately and this is now clearly illustrated in Supplementary Fig. 13a,b.

We have revised large parts of the computational part, including two important subsections, '*Analysis of protein expression with age and statistical significance calculation (TMT)*' (page 10) and '*Statistical methods to determine changes between cell populations related to Fig. 2c (LF)*' (page 10). The detailed technical explanations for TMT- and LF-quantification are provided in the sub-sections '*Data analysis for TMT based quantification*' (page 7) and '*Data analysis for label-free (LF) quantification*' (page 8). We added a "(TMT)" or "(LF)" label to the title of the sub-sections to indicate which dataset and quantification are described.

(ii) We agree with Reviewer #3, that the detailed information on the internal standard was difficult to find in the Methods section and was not detailed enough. We have therefore made a separate paragraph with the header '*Preparation of internal standards for proteomics analysis*' (page 3) and added more details.

(iii) We have provided a detailed description of the statistical tools used to determine which proteins show statistically significant age-associated changes (TMT-quantifications) in the sub-section '*Analysis of protein expression with age and statistical significance calculation (TMT)*' (page 10). The section includes an explanation on:

a) which proteins were considered as successfully quantified. In this analysis, we considered only proteins that were consistently measured in at least 15% of the donors (for a given cell population).

b) how significant age-associated changes in protein abundance were characterized. Briefly, Spearman correlation was applied on TMT-ratios to derive age-dependencies (correlation coefficients), with the respective p-values. We listed all p-values in Supplementary Table 5. For subsequent analyses, we considered that p-values < 0.05 were significant. Only in specific cases (and only for information), we also mentioned p-values comprised between 0.05 and 0.1 (e.g. Fig. 4, 6).

For differences between the cell populations (LF-scores), we added an entirely new section '*Statistical methods to determine changes between cell populations related to Fig. 2c (LF)*'. It contains a detailed description of the statistical analysis performed to characterize which pathways are significantly affected in their stoichiometry between cell populations (page 10).

Next, the normalization process needs to be carefully explained. If there are an uneven number of samples from a given cell type at a certain age, which appears to be the case from supplemental table 3, for example, how do the authors account for changes that may appear due to under-sampling since they have less material to analyze? Given the nature of proteomics if the same number of cells and/or total protein starting material is not used, changes from one cell type to another could possibly be explained by having more total material for analysis from one sample to another. The authors need to clearly explain how they corrected for this potential confounding issue.

Reviewer #3 expressed concern that the normalization procedure was inadequately delineated in the Methods section. We provided a detailed processing scheme where 'Normalization' is described in the first section (a) (see Supplementary Fig. 13). We noticed that TMT- and LF-quantification could be easily confused in the previous version, and we therefore introduced a more clear-cut structure with respective captions throughout the entire Methods section. The normalization procedure is now described in detail separately for TMT-quantification (page 8), as well as for LF-quantification (page 9) in two new sections called "Summary of the normalization steps of the TMT data" and "Summary of the normalization steps of the LF data".

Regarding the size of the material for proteomics analysis, we had standardized our procedure and were able to account for experimental variation. Several quality control measures were taken, which are described in a section called 'Quality assessment and sample inclusion in the proteomics analysis' (page 9). We would like to clarify the procedures here:

(i) The cell pellets were similar in size. The cell samples were obtained after FACS sorting and we obtained a good estimate of the cell numbers. We always processed pellets that contained a similar number of cells. For MSC, MON, LYM and GRA these were ~500,000 cells; for ERP ~1,000,000 cells, and for HPC ~100,000 cells.

(ii) For the proteomics sample preparation (tryptic digestion and TMT labeling), we used standardized protocols (describe in the Methods section page 3-4). These steps were carefully controlled. The efficiency of tryptic digestion was estimated by the number of missed-cleaved peptides and the efficiency of TMT labeling by the ratio of completely versus partially, non-labelled peptides (Supplementary Fig. 2). Samples of bad quality and/or deviating from the norm (PCA analysis) were excluded from further analysis (Supplementary Fig. 2c).

(iii) We carefully controlled the amounts of sample materials for mass spectrometry analysis (1 to 1 ratio of all TMT labeled samples and internal standard). We first performed a short test run with equal volumes of all (5) TMT-labeled samples and the corresponding TMT-labeled internal standard. The results were then used to adjust the volumes to account for possible differences in sample material in the final mixing. We also added a step of *in silico* normalization and adjusted the median intensities of the five samples (TMT channels) to the one of the internal standard.

(iv) Normalization for the LF-quantification is more delicate, as the different cell populations, which we measured here, have different sizes, and subcellular structures. Therefore, we normalized for median protein abundance and corrected for the overall number of samples available per cell population, which means that we provided relative protein abundance (described in detail now in sub-section 'Summary of the normalization steps of the LF data' on page 9). In addition, we focused our downstream LF-analyses on the comparison of the relative stoichiometry of different pathways, or cellular processes across cell types. This means that the changes we report are changes in stoichiometry, i.e. normalized for a pathway (Figure 2c).

The last major issue for the manuscript is assuming all of the above issues can be clarified, the manuscript contains proteomics data alone and simply compares the results to prior literature. It is unclear what the major new knowledge is here given the extensive literature on ageing. The mitochondria play an important role in ageing (Fikel T, Nature Medicine 2015), for example, but little description of the mitochondria changes, if any, are described here.

We agree with Reviewer #3 that mitochondria play an important role in ageing. Accordingly, we have added a brief description of the changes we observed in this organelle (page 6, last paragraph). The significant changes (p-value <

0.05) in mitochondrial associated proteins are compatible with the results reported in the literature as reviewed by Finkel et al. in 2015¹⁴.

In addition to recapitulating results known previously from murine studies our study has provided novel knowledge on (a) the alterations in central carbon metabolism in ageing HPC; and (b) the complementary decrease in levels of adhesive molecules and respective ligands in the HPC and MSC. In the revised version, we have significantly expanded the analysis of the major novelties listed above (see below).

Also, there is no validation of biological insights using alternative approaches. Comparing their results to existing transcriptomics and epigenetic data (Refs 17 and 18) could be insightful. Carrying out assays to demonstrate changes in glycolysis of cells would be valuable. Even if the protein levels of central carbon metabolism are altered, no data is presented to determine what effect this has on central carbon metabolism itself. It is assumed that carbon metabolites would be altered, but not evidence is provided.

We appreciate the constructive comment of Reviewer #3 and that comparisons of the proteomic findings using alternative approaches to transcriptomics data could be insightful; and validating the alterations in central carbon metabolism upon ageing might provide further evidence. Concerning the already published transcriptomics and epigenetics data mentioned by Reviewer #3, they are limited to one cell population, the MSCs, and arise from a largely different cohort of human subjects. We believe that the scope of these published data are inadequate to validate proteomics data derived from a broad collection of human samples and cell populations.

We have in the meantime performed additional validation experiments and significantly expanded the analyses of the major novelties.

(1) As suggested by the Reviewer #3, we have confirmed the results of our investigation by adding results from an orthogonal methodology, such as RNA-seq. Most importantly, these new transcriptomic analyses were conducted on samples FACS sorted on the same day as the ones used for the proteomics analyses (i.e. originate from the same cohort of volunteers). Overall, 65 RNA-seq analyses (11 HPCs, 13 LYM, 12 MON, 9 GRA, 12 ERP, 8 MSC) implying 13 donors were performed. For all cell populations, and a number of young and old donors, we could overlay the transcriptomic results with the information gathered from the proteome level. In general, we observed genes that were up-regulated at the protein level (p -value < 0.05) to exhibit a significantly higher transcript fold-change than for genes that are down-regulated at the protein level (Supplementary Fig. 9b). The transcriptomics data support the major novelties that we describe in this manuscript (Supplementary Fig. 9c). This is described on page 6 (first paragraph) of the main text.

(2) We have also performed additional experiments to address whether the increase in glycolytic enzymes in old HPCs could be a direct consequence of the lineage skewing of the CD34+ cells towards myeloid differentiation. For this purpose, we analyzed the transcriptome of 519 single-sorted HPCs originating from young ($n=2$) and old ($n=2$) human subjects. Based on the abundance levels of the mRNA markers linked with lymphoid or myeloid differentiation as described in Figure 5 (and that we could consistently measure in single cells), we categorized each of the 519 HPCs (Supplementary Figure 12a,b,c). These results have demonstrated that the mRNA levels of age-increased glycolytic enzymes were expressed at significantly higher levels in myeloid-primed than in lymphoid-primed HPCs, whereas age-unaffected enzymes have similar mRNA levels in both subsets (Figure 5d,e and Supplementary Figure 12d,e). Remarkably, the age-dependent increase in glycolytic enzymes was most prominent in the myeloid-primed subset of HPCs. Thus, the lineage skewing of the HPCs towards myeloid differentiation observed upon ageing may account, at least in part, for the increase in abundance of glycolytic enzymes. These new data are now described on page 9 (third paragraph) of the main text.

(3) We performed metabolomics analyses of HPCs ($n=10$, age: 21-69 year old). This required the development of a new method - based on the derivatization of sugar-phosphates with 3-amino-9-methylcarbazole – which was sensitive enough to detect metabolites in only ~100,000 cells with the limit of detection being in the low femtomolar

range (the method is described in detail on page 5-6 and page 12 in the Methods section). The following metabolites were measured: glucose 6-phosphate, ribose 5-phosphate, ribulose 5-phosphate, erythrose 4-phosphate, dihydroxyacetone-phosphate / glyceraldehyde 3-phosphate, and mannose 6-phosphate / galactose 6-phosphate. The data showed a tendency for ribose 5-phosphate and ribulose 5-phosphate (in the pentose phosphate pathway) to accumulate in aged HPCs, whereas the other analyzed metabolites showed no age correlation. This is compatible with the notion that the age-associated increase in abundance of enzymes catalyzing the rate-limiting steps of the upper part of glycolysis has led to increased fluxes through the pentose phosphate and nucleic acid synthesis pathways. The results are added to Fig. 4 and are described on page 7 ('Results') and page 10-11 ('Discussion') in the main text.

References

1. Dominici M, *et al.* Minimal criteria for defining multipotent mesenchymal stromal cells. The International Society for Cellular Therapy position statement. *Cytotherapy* **8**, 315-317 (2006).
2. Wagner W, *et al.* Replicative Senescence of Mesenchymal Stem Cells: A Continuous and Organized Process. *PLoS One* **3**, (2008).
3. Wagner WG, *et al.* Molecular and secretory profiles of human mesenchymal stromal cells and their abilities to maintain primitive hematopoietic progenitors. *Stem Cells* **25**, 2638-2647 (2007).
4. Wagner W, *et al.* Aging and Replicative Senescence Have Related Effects on Human Stem and Progenitor Cells. *PLoS One* **4**, (2009).
5. Bork S, *et al.* DNA methylation pattern changes upon long-term culture and aging of human mesenchymal stromal cells. *Aging Cell* **9**, 54-63 (2010).
6. Bocker MT, Hellwig I, Breiling A, Eckstein V, Ho AD, Lyko F. Genome-wide promoter DNA methylation dynamics of human hematopoietic progenitor cells during differentiation and aging. *Blood* **117**, E182-E189 (2011).
7. Uhlen M, *et al.* Tissue-based map of the human proteome. *Science* **347**, (2015).
8. Kuranda K, *et al.* Age-related changes in human hematopoietic stem/progenitor cells. *Aging Cell* **10**, 542-546 (2011).
9. Pang WW, *et al.* Human bone marrow hematopoietic stem cells are increased in frequency and myeloid-biased with age. *Proc Natl Acad Sci U S A* **108**, 20012-20017 (2011).
10. Mendez-Ferrer S, *et al.* Mesenchymal and haematopoietic stem cells form a unique bone marrow niche. *Nature* **466**, 829-U859 (2010).
11. Xie L, Zeng X, Hu J, Chen QM. Characterization of Nestin, a Selective Marker for Bone Marrow Derived Mesenchymal Stem Cells. *Stem Cells Int*, (2015).
12. Pinho S, *et al.* PDGFR alpha and CD51 mark human Nestin(+) sphere-forming mesenchymal stem cells capable of hematopoietic progenitor cell expansion. *J Exp Med* **210**, 1351-1367 (2013).
13. Isern J, *et al.* Self-Renewing Human Bone Marrow Mesospheres Promote Hematopoietic Stem Cell Expansion. *Cell Reports* **3**, 1714-1724 (2013).
14. Finkel T. The metabolic regulation of aging. *Nat Med* **21**, 1416-1423 (2015).

Reviewers' Comments:

Reviewer #1:

Remarks to the Author:

The authors have done a commendable job in responding to the concerns raised by reviewers. The paper is now markedly more valuable as a resource and is likely to be a highly cited reference in the field.

Reviewer #2:

Remarks to the Author:

The authors have adequately addressed the reviewers' comments.

Reviewer #3:

Remarks to the Author:

Outstanding revision, I fully support publication of this work.

Reviewer #1 (Remarks to the Author):

The authors have done a commendable job in responding to the concerns raised by reviewers. The paper is now markedly more valuable as a resource and is likely to be a highly cited reference in the field.

We appreciate the comment of Reviewer #1 that "...valuable as a resource and is likely to be a highly cited reference in the field"

--

Reviewer #2 (Remarks to the Author):

The authors have adequately addressed the reviewers' comments.

We thank Reviewer #2 for the positive feedback.

--

Reviewer #3 (Remarks to the Author):

Outstanding revision, I fully support publication of this work.

We thank Reviewer #3 for the enthusiastic support.